# Massive reduction of RyR1 in muscle spindles of mice carrying recessive *Ryr1* mutations alters proprioception and causes scoliosis

Alexis Ruiz[1] , Sofia Benucci[1], Hervé Meier[1], Georg Schultz[2] , Katarzyna Buczak[3], Christoph Handschin[4] , Rodrigo C. G. Pena[5] , Susan Treves[1,6] and Francesco Zorzato[1,6] 

[1]*Neuromuscular Research Group, Departments of Neurology and Biomedicine, Basel University Hospital, Basel, Switzerland*
[2]*Biomaterial Science Center, Department of Biomedical Engineering, University of Basel, Allschwil, Switzerland*
[3]*Proteomics Core Facility, Biozentrum, University of Basel, Basel, Switzerland*
[4]*Biozentrum, University of Basel, Basel, Switzerland*
[5]*Center for Data Analytics (CeDA), University of Basel, Basel, Switzerland*
[6]*Department of Life Science and Biotechnology, University of Ferrara, Ferrara, Italy*

Handling Editors: Karyn Hamilton & Nikki Jernigan

The peer review history is available in the Supporting Information section of this article (https://doi.org/10.1113/JP287832#support-information-section).

The Journal of Physiology

**Abstract figure legend** Intrafusal muscles contained within muscle spindles are endowed with ryanodie receptor 1 (RyR1) calcium channels and participate in proprioceptor function. Mutations in RyR1 linked to severe RYR1-congenital myopathies affect calcium release from both extrafusal as well as intrafusal muscles. Altered calcium signaling via mutant RyR1 calcium channels affect proprioceptor function and may be linked to skeletal abnormalities.

S. Treves and F. Zorzato contributed equally to this work.

This article was first published as a preprint. Ruiz A, Benucci S, Meier H, Schultz G, Buczak K, Handschin C, Pena RCG, Treves S, Zorzato F. 2024. Massive reduction of RyR1 in muscle spindles of mice carrying recessive Ryr1 mutations alters proprioception and causes scoliosis. bioRxiv. https://doi.org/10.1101/2024.08.09.607317

The Journal of Physiology

**Abstract** Muscle spindles are stretch receptors lying deep within the muscle belly involved in detecting changes in muscle length and playing a fundamental role in motor control, posture and synchronized gait. They are made up of an external capsule surrounding three to five intrafusal muscle fibres and a nuclear bag complex. Dysfunction of muscle spindles leads to abnormal proprioceptor function, which has been linked to aberrant bone and cartilage development, scoliosis, kyphosis and joint contractures. *RYR1*, the gene encoding the calcium release channel of the sarcoplasmic reticulum, is the most common target of mutations linked to human congenital myopathies, a condition often accompanied by skeleton alterations and joint contractures. So far, the link between *RYR1* mutations, altered muscle spindles and skeletal defects has not been investigated. To this end, we investigated heterozygous mice carrying recessive *Ryr1* mutations isogenic to those present in a severely affected child. Here, we show that: (i) the RyR1 protein localizes to the polar regions of intrafusal fibres and exhibits a doubled row distribution pattern, typical for junctional sarcoplasmic reticulum proteins; (ii) muscle spindles of compound heterozygous mice show structural defects; and (iii) in intrafusal muscle fibres from dHT mice, RyR1-mediated $Ca^{2+}$ release is significantly impaired and RyR1 protein content is reduced by 54%. These results support the hypothesis that *Ryr1* mutations not only affect the function of extrafusal muscles, but also might affect that of intrafusal muscles. The latter may be one of the underlying causes of skeletal abnormalities seen in patients affected by recessive *RYR1* mutations.

(Received 10 October 2024; accepted after revision 1 September 2025; first published online 29 September 2024)

**Corresponding author** S. Treves: LAB 408 ZLF, Hebelstrasse 20, 4031 Basel, Switzerland. Email: susan.treves@unibas.ch.

**Key points**

- Dysfunction of muscle spindles leads to abnormal proprioceptor function, which can cause abnormal bone and cartilage development, scoliosis, kyphosis and joint contractures.
- Muscle spindles contain intrafusal muscle fibres, which are made up of nuclear bag and nuclear chain fibres; the polar regions of intrafusal muscle fibres contain contractile filaments and sarcotubular membranes.
- Patients with *RYR1* mutations often present skeleton alterations and joint contractures from birth.
- Using the dHT mouse model carrying recessive *Ryr1* mutations, we investigated whether muscle spindles from dHT mice show structural defects and altered RyR1 content and function.
- Our results support the hypothesis that *RYR1* mutations also affect the function of intrafusal muscles and this may be one of the underlying causes of the skeletal abnormalities seen in patients.

## Introduction

Dysregulation of calcium homeostasis in skeletal muscles plays an important role in the pathogenesis of a number of neuromuscular disorders (Jungbluth et al., 2018; Treves et al., 2017). Disruption of the calcium signals can be brought about by mutations in a number of genes encoding key proteins involved in excitation-contraction coupling (ECC) including the ryanodine receptor type 1 (RyR1), voltage-dependent calcium channels

**Alexis Ruiz**, originally from Venezuela, began his scientific journey studying muscle biophysics and metabolism. More recently, he has shown how *Ryr1* mutations affect skeletal muscle function and may contribute to skeleton malformations. Such insights can help explain many phenotypic characteristics of patients with severe RYR1-myopathies. A highlight of his career has been contributing to the discovery of potential treatments for muscle diseases. Looking ahead, his aims to deepen our understanding of skeletal muscle function and tackle the challenges of treating congenital muscle disorders. He remains driven by a passion for uncovering the complexities of muscle physiology.

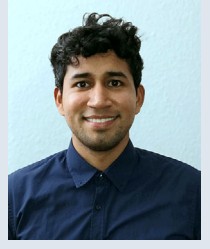

(dihydropyridine receptor, DHPR), STAC3, Stim1, Orai1 and calsequestrin 1 (CASQ1) (Amburgey et al., 2013; Jungbluth et al., 2018; Lawal et al., 2020; Treves et al., 2017; Endo et al., 2015; Horstick et al., 2013; Morin et al., 2020; Schartner et al., 2017; Semplicini et al., 2018). Depolarization of the sarcolemma is sensed by the DHPR, which delivers the signal to release the $Ca^{2+}$ stored in the sarcoplasmic reticulum (SR) terminal cisternae by opening the RyR1 calcium release channel, leading to muscle contraction by the process known as ECC. The re-uptake of cytosolic calcium by the SR CaATPase induces muscle relaxation (Endo, 1977; Fleischer & Inui, 1988; Rios & Pizarro, 1991). Mutations in *RYR1*, the gene encoding RyR1, underlie several neuromuscular disorders, including malignant hyperthermia (MH; MIM #145600), central core disease (CCD; MIM #11700), specific forms of multi-minicore disease and centronuclear myopathy (MmD, CNM; MIM # 255320) (Amburgey et al., 2013; Jungbluth et al., 2018; Lawal et al., 2020; Treves et al., 2017).

A great deal of experimental data has shown that *RYR1* mutations result in different types of channel defects ranging from alterations of regulatory mechanisms of channel activity to deficits of RyR1 expression (Jungbluth et al., 2018; Treves et al., 2008; Zhou et al., 2007; Zhou et al., 2013). The latter is characteristic of recessive mutations linked to congenital myopathies such as MmD and CNM. The clinical presentation of patients harboring recessive *RYR1* mutations usually occurs in infancy or early childhood and is characterized by motor developmental delays, hypotonia, proximal muscle weakness and muscle stiffness. Additional symptoms and phenotypes include involvement of extraocular muscles leading to ophthalmoplegia and/or ptosis and involvement of respiratory muscles, often requiring assisted ventilation (Amburgey et al., 2013; Jungbluth et al., 2018; Lawal et al., 2020; Jungbluth et al., 2005; Klein et al., 2012; Sarkozy et al., 2023). Of importance, many patients present a number of skeletal abnormalities at birth, including scoliosis and congenital dislocation of the hip, kyphosis, clubfoot (talipes equinovarus), flattening of the arch of the foot (flatfoot or pes planus) or an abnormally high arch of the foot (pes cavus) (Amburgey et al., 2013; Sarkozy et al., 2023). These observations are particularly intriguing and point to a potential role of the RyR1 in the correct development and maintenance of the musculoskeletal system, a hypothesis sustained by the observation that embryos of mice and birds depleted of RyR1, or carrying homozygous mutations in the *Ryr1* have skeletal abnormalities including kyphosis, excavated sternum, abnormal head positioning, spreading of the toes and crossing of the legs at the tibiotarsal joints (Airey et al., 1993; Chelu et al., 2005; Takeshima et al., 1994; Sittmann & Craig, 1967).

Fetal skeletal muscle movements appear to be relevant for the appropriate development of the skeletal system and bone morphogenesis as demonstrated by the following observations: (i) fetal akinesia or insufficient fetal movements results in abnormal joint development and joint contractures (Davis & Kalousek, 1988; Hall, 2014); (ii) the presence of *RYR1* mutations is also linked to fetal akinesia syndrome (McKie et al., 2014; Romero et al., 2003); and (iii) muscular dysgenesis mice carrying mutations in the $\alpha$1s of the DHPR and lacking ECC, exhibit arrest of chondrocyte proliferation in the developing bone eminences, resulting in their loss (Powell, 1990).

Nevertheless, studies on the factors involved in appropriate musculoskeletal development are in their infancy. One recently identified player is the mechanosensitive ion channel Piezo2. In humans, mutations in *PIEZO2* lead to scoliosis, hip dislocation, joint contracture, arthrogryposis, respiratory distress and muscle weakness (Alper, 2017; Assaraf et al., 2020; Chelser et al., 2016; Kröger & Watkins, 2021; McMillin et al., 2014). Piezo2 is expressed in muscle spindles and Golgi tendon organs, comprising two proprioceptive mechanosensory organs (Woo et al., 2015). Muscle spindles do not significantly contribute to the generation of muscle force but are stretch detectors. When a muscle is stretched, the change in length is transmitted to the muscle spindles that lie deep within the belly of the muscle; muscle spindles then convey the information concerning muscle stretch as well as the speed of stretching via sensory neurons to the central nervous system, which responds by adjusting the response of agonistic and antagonistic muscle groups (Banks et al., 2005). Each muscle spindle contains approximately three to five intrafusal muscle fibres (in mouse) made up of nuclear bag and nuclear chain fibres; the polar region of these specialized muscle fibres also contains contractile filaments, as well as sarcotubular membranes, including longitudinal tubules and triads similar to those present in extrafusal fibres (Ovalle, 1972; Ovalle, 1976). Triads are endowed with calcium release units (CRU), and are formed by two SR terminal cisternae closely opposed to a central tubule oriented transversally with respect to the longitudinal axis of the myofibrils (Franzini-Armstrong & Jorgensen, 1994). In muscle spindles, activation of the CRU in response to gamma motor neuron stimulation causes contraction of the polar region of intrafusal fibres, an event that would offset the slackening of intrafusal fibres during the shortening of extrafusal fibres, thus providing a continuous adjustment of muscle spindle afferent firing during normal locomotor activity. On the basis of these findings, we hypothesize that the presence of mutated RyR1 in the CRU may affect the intrafusal fibres of muscle spindles by interfering with the adjustment of gamma motor neuron mediated intrafusal fibre contraction. This could result in proprioceptive signalling abnormalities in paravertebral muscles,

which in turn may induce musculoskeletal deformities, features that have been observed in many patients with *RYR1* mutations (Amburgey et al., 2013; Lawal et al., 2020; Sarkozy et al., 2023).

Here, we investigated muscle spindle structure and gait properties in transgenic mice carrying the compound heterozygous *Ryr1* mutations RyR1p.Q1970fsX16+p.A4329D (referable as dHT) a model for recessive *Ryr1* myopathies (Elbaz et al., 2019a). We also investigated muscle spindle structure and gait properties of two additional mouse models carrying mutations in *Ryr1*. In particular, we examined Ex36 mutant mice, carrying the heterozygous RyR1p.Q1970fsX16 mutation, leading to the expression of a single wild-type (WT) allele (Elbaz et al., 2019b), and of Homozygous mice (HO mice), carrying the homozygous RyR p.F4976L mutation (Benucci et al., 2024). These two mouse models exhibit a milder skeletal muscle phenotype compared to the dHT model, as well as a smaller reduction of RyR1 expression and content (Benucci et al., 2024; Elbaz et al., 2019b). Our results show that dHT mice, but not Ex36 and HO mice, display changes in the histological appearance of muscle spindles and skeleton including: (i) alterations of the structure of the muscle spindles with an enlarged gap between the muscle spindle and extrafusal fibres; (ii) alterations of gait properties; and (iiii) skeleton abnormalities, in particular the presence of scoliosis as indicated by a 20% increase of the Cobb angle. High resolution proteomic analysis of muscle spindle samples collected by laser capture micro-dissection from WT, Ex36 and dHT mice confirms the presence of the RyR1 protein in intrafusal fibres and its strong reduction in content in dHT mice. Additionally, ECC is impaired as shown by the reduced RyR1-mediated $Ca^{2+}$ release in muscle spindles from dHT mice compared to WT and Ex36 littermates.

In conclusion, our study shows that the presence of recessive *Ryr1* mutations might negatively influence not only the function of extrafusal muscles, but also that of muscle spindle intrafusal muscle fibres.

## Methods

### Ethical approval

We adhere to *J Physiol* policies regarding animal experiments. All experimental procedures on mice were approved by the Kantonal Veterinary Authority of Basel Stadt (BS Kantonales Veterinäramt Permit numbers 1728 and 2115). All experiments were carried out in accordance with the Basel Stadt Kantonal Veterinary guidelines and regulations. All mice were housed in cages (12:12 h light/dark photocycle; lights on 05.00 h) and had free access to food and water. For muscle isolation, mice were deeply anesthetized by an I.P. injection of pentobarbital;

once the muscles were isolated, mice were killed by an overdose of sodium pentobarbital (150–200 mg kg$^{-1}$) administered I.P. Death of the mice was confirmed by the absence of breathing and the absence of a heartbeat.

### Animal models

The following mouse models were used: dHT carrying the compound heterozygous RyR1p.Q1970fsX+p.A4329D mutations (Elbaz et al., 2019a) isogenic to those identified in a severely affected child (patient 28, Zhou et al., 2007; patient 56, Klein et al., 2012), Ex36 carrying the heterozygous RyR1p.Q1970fsX mutation (Elbaz et al., 2019b), HO carrying the homozygous RyR1p.F4976L mutation (Benucci et al., 2024), and their WT littermates, as well as B6-Cg-Tg (Thy1-EYFP) mice from Jackson Laboratories (Bar Harbor, ME, USA). The latter were also bred with dHT and Ex36 transgenic mice for confocal immunohistochemical analysis and calcium imaging on muscle spindles All experiments were conducted on 11–12-week-old male mice unless otherwise stated

### Mouse genotyping

PCR amplification of *Ryr1* exons 36, 91 and 104 was performed as previously described (Benucci et al., 2024; Elbaz et al., 2019a) on genomic DNA of WT, Ex 36, RyR1Q1970fsX16+A4329D and HO (RyR1F4976L) transgenic mice. B6-Cg-Tg (Thy1-EYFP) genotyping was carried out as described (https://www.jax.org/Protocol?stockNumber=003709&protocolID=32064) using Probe Based quantitative PCR (qPCR) and an Internal Positive Control (IPC) as a reference, except that the qPCR kit from Promega (Madison, WI, USA; catalog. no. A6102) was used. Each primer set was used in combination with a probe for fluorescence quantification during the qPCR amplification reaction. The complete list of primers used for genotyping is given in Table 1.

### Determination of spindle diameter by hematoxylin and eosin staining

Soleus muscles were isolated from 12-week-old male mice, embedded in PolyFreeze (Sigma, St Louis, MO, USA; catalog. no. P0091), deep-frozen in 2-methylbutane (Sigma; catalog. no. M32631) and stored at $-80°C$. Muscles were then cut in 10 µm thick sections using a cryostat (Leica, Wetzlar, Germany; CM1950) and stored at $-80°C$. Samples were stained with hematoxylin and eosin (H&E) using a commercial staining kit (Abcam, Cambridge, UK; catalog. no. 245 880). Sections were visualized using a 40× Plan Apo λ objective (N.A. = 0.95) using an Eclipse Ti2 widefield microscope (Nikon, Tokyo, Japan). The diameter of muscle spindles (distance of

**Table 1. Sequence of primers used for genotyping**

| Gene amplification details | Primer sequence | |
|---|---|---|
| | Forward | Reverse |
| *Ryr1* Ex36 | 5′-TGC TGG CTT CAG AGT GAT CG-3′ | 5′-CGA GGG AAG TTG AGG TTG GG-3′ |
| *Ryr1* Ex91 | 5′-GAG ATG TTC GTG AGT TTC TGC GAG G-3′ | 5′-TGA GGG TTG TTC TTG GTG TAT TTG G-3′ |
| *Ryr1* Ex104 | 5′-GAC CAA CAA GAG CAA GTG AAG G -3′ | 5′-GTC TAA ACC CTA CCC ACA CG-3′ |
| Thy1-GFP | 5′-CAC AGA ATC CAA GTC GGA ACT-3′ | 5′-AAC AGC TCC TCG CCC TTG-3′ |
| IPC Thy1-GFP | 5′-CAC GTG GGC TCC AGC ATT-3′ | 5′-TCA CCA GTC ATT TCT GCC TTT G-3′ |
| TG_Thy1_probe | 5′-CAC CTA GAG GAT CTC GAG GGA TCC-3′ *5′ modification*: FAM *3′ modification*: TAMRA | |
| IPC_Thy1_probe | 5′-CCA ATG GTC GGG CAC TGC TCA A-3′ *5′ modification*: HEX *3′ modification*: TAMRA | |

parallel tangents at opposing borders of the fibre) was calculated from the central region of the muscle spindle (equatorial region), which was recognized after scrolling through the individual images of the stack (Gartych et al., 2021). Images were analysed using Fiji software (NIH, Bethesda, MD, USA).

## Immunohistochemical analysis of muscle spindles

Twelve-week-old male or female mice were deeply anesthetized by an I.P. injection of pentobarbital and perfused with 4% paraformaldehyde for fixation. Fixed extensor digitorum longus (EDL) and soleus muscles were isolated and embedded in PolyFreeze medium (Sigma; catalog. no. P0091), frozen in cold hexane and stored at –80°C until use. Longitudinal 25 μm sections were obtained using a cryostat (Leica; CM1950) and processed as described previously (Watkins et al., 2023). For staining, sections were first rehydrated in phosphate-buffered saline (PBS) for 10 min. Antigen retrieval was performed by steaming in the immunohistochemistry antigen retrieval buffer (Thermo Fisher Scientific, Waltham, MA, USA; catalog. no. 00-4955-58). Blocking was performed using a blocking solution (western blocking reagent Roche, Basel, Switzerland; catalog. no. 11921673001) containing 0.15% Triton X-100 and 1% bovine serum albumin in 300 mM glycine for 1 h at room temperature. Subsequently, sections were incubated overnight at 4°C with the following primary antibodies: rabbit D4E1 monoclonal anti-RyR1 (Cell Signaling Technology, Danvers, MA, USA; catalog. no. 8153; 1:150 dilution in PBS), chicken anti-GFP to visualize YFP-Thy1 (Abcam; catalog. no. ab13970; 1:500 dilution in 1% bovine serum albumin in PBS). The next morning, samples were washed at room temperature three times for 10 min each with 0.1% Tween in PBS (PBS-T) and then incubated for 1 h at room temperature with the following secondary anti-

bodies: goat anti-rabbit Alexa Fluor 647 (Thermo Fisher Scientific; catalog. no. A21245; 1:1000 dilution in PBS-T) and goat anti-chicken Alexa Fluor 488 (Thermo Fisher Scientific; catalog. no. A11039; 1:1000 dilution in PBS-T). Samples were washed twice for 10 min each with PBS-T and once for 10 min with PBS and then incubated with 4′,6-diamidine-2′-phenylindole dihydrochloride (DAPI) (Sigma; catalog. no. D9542; 1:500 dilution in PBS) for 15 min at room temperature. Finally, samples were washed twice for 10 min each with PBS and once for 10 min, mounted in Fluoromount G (Thermo Fisher Scientific; catalog. no. 00-4958-02) and visualized using a confocal AxR microscope (Nikon) equipped with a 40× Plan APO objective (N.A. = 095).

## Laser capture microdissection (LCM) and mass spectrometry

Soleus and EDL muscles isolated from 11–13-week-old male mice were embedded in OCT (Sigma), deep-frozen in 2-methylbutane (Sigma), cut into 25 μm thick sections as described above and stored at –80°C. Spindles identified in consecutive cross-sections were isolated by laser capture microdissection as described below, using a PALM Robot-microbeam system (P.A.L.M. Microlaser Technologies AG, Jena, Germany) equipped with the PALM RoboSoftware version 1.2 (P.A.L.M. Microlaser Technologies AG) and mounted on a Zeiss Axiovert 200 M microscope (Carl Zeiss, Oberkochen, Germany). Muscle spindles identified within a muscle section, were encircled in an 'region of interest' and dissected out of the tissue using the cutting and catapulting function, RoboLPC. The catapulted material was collected in the cap of a 200 μL Thermo-Tube (ABgene, Epsom, UK) containing 20 μL of lysis buffer (5% SDS; 50 mM triethylammonium bicarbonate) as shown in Fig. 1*A*. Twenty-five micro-

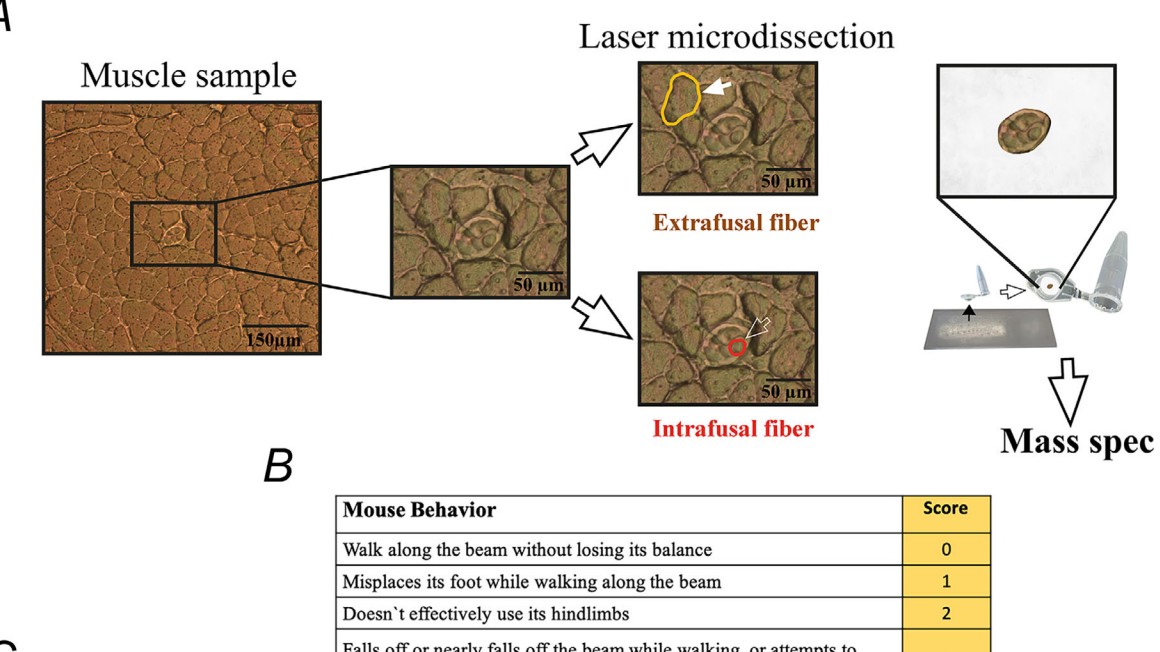

*A*

Muscle sample

Laser microdissection

50 µm

**Extrafusal fiber**

50 µm

50 µm

**Intrafusal fiber**

150µm

**Mass spec**

*B*

| Mouse Behavior | Score |
|---|---|
| Walk along the beam without losing its balance | 0 |
| Misplaces its foot while walking along the beam | 1 |
| Doesn`t effectively use its hindlimbs | 2 |
| Falls off or nearly falls off the beam while walking, or attempts to lower itself, or refuses to move | 3 |

*C*

**Step Cycle (s)** is the time in seconds between two consecutive initial contactts of the same paw:

Step Cycle = Stand + Swing

The figure below graphically depicts Step Cycle, Stand and Swing.

RH — Step Cycle

RH

Stand    Swing

**Duty Cycle**

Duty Cycle (%) expressed Stand as a percentage of Step Cycle

$$Duty\ Cycle = \frac{Stand}{Stand + Swing} \times 100\ \%$$

Right Front
Right Hind
Left Front
Left Hind

BOS Hindpaws

BOS Forepaws

Left Front

Left Front

Stance    Swing    Step cycle

**Dual Stance** is the duration (in seconds) of ground contact for both hind paws simultaneusly (Coulthand et al. 2002, 2003). Dual Stance is used for gait analysis.

**The Initial Dual Stance** is the fist time in a step cycle of a hind paw that the contralateral hind paw also makers contact with the glass.

**Terminal Dual Stance,** is the secons step in a Step Cycle of a hind paw that the contralateral hind paw also makes contact with the glass plate.
Stand is the duration (s) of contact of a paw with the glass plate.

**Single and Dual Stance** are calculated if the contralateral hind paw has been placed on the glass plate before and after a step cycle of the hind paw.

Right Front
Right Hind
Left Front
Left Hind

**Timing view. 1.** Step cycling of the right hind paw. **2.-** initial Dual Stance od RH and LH. **3.-** Single Stance of RH **4.-** Terminal Dual Stance of RH and LH.

**Figure 1. Description of *in vivo* muscle spindle assessment parameters and schematic representation of the laser capture microscopy**
*A*, areas of interest identified within a muscle section were collected by laser capture microdissection (LCM) and subsequently processed for quantitative proteomic analysis as described in the Methods. *B*, Beam walk scoring parameters. *C*, main gait parameters and their explanation, measured by the CatWalk system.

dissected sections from the polar area of the muscle spindle were isolated per mouse.

For mass spectrometry, 10 mm Tris 2-carboxyethyl phosphine was added to each sample, followed by heating to 95°C for 10 min and sonication (Bioruptor, 10 cycles, 30 s on/off; Diagenode, Seraing, Belgium). After equilibrating to room temperature, iodoacetamide was added and the samples were then incubated at room temperature (protected from light) for 30 min. Proteins were then purified and digested following the modified SP3 protocol (Hughes et al., 2019). In brief, Speed Beads (#45152105050250 and #65152105050250; GE Healthcare, Chicago, IL, USA) were mixed 1:1, rinsed with water and diluted to the 8 μg μL$^{-1}$ stock solution. Samples were adjusted to the final volume of 25 μL and 5 μL of the bead stock solution was added to the samples. Proteins were bound to the beads by addition of 30 μL of 100% acetonitrile to the samples (to the final concentration of 50%), which were then incubated for 8 min at room temperature with gentle agitation (200 rpm). After, samples were placed on a magnetic rack and incubated for 5 min. Supernatants were removed and discarded. The beads were washed twice with 200 μL of 70% (v/v) ethanol and once with 200 μL of 100% acetonitrile. Samples were placed off the magnetic rack and 10 μL of digestion mix (5 ng μL$^{-1}$ trypsin in 0.02% *n*-dodecyl-beta-maltoside, 50 mm triethylammonium bicarbonate) was added to them. Digestion was allowed to proceed for 12 h at 37°C. After digestion samples were placed back on the magnetic and incubated for 5 min. Supernatants containing peptides were collected and dried under vacuum.

Dried peptides were dissolved in 10 μL of 0.1% formic acid and 4 μl of the sample was subjected to liquid chromatography-tandem mass spectrometry analysis using an Orbitrap Eclipse Tribrid Mass Spectrometer fitted with an Ultimate 3000 nano-LC (both Thermo Fisher Scientific) and a custom-made column heater set to 60°C using block randomization.

Peptides were resolved using a reverse phase-HPLC column (75 μm × 30 cm) packed in-house with C18 resin (ReproSil-Pur C18-AQ, 1.9 μm resin; Dr Maisch GmbH, Ammerbuch-Entringen, Germany) at a flow rate of 0.3 μL min$^{-1}$. The following gradient was used for peptide separation: from 2% B to 12% B over 5 min to 30% B over 40 min to 50% B over 15 min to 95% B over 2 min followed by 11 min at 95% B. Buffer A was 0.1% formic acid in water and buffer B was 80% acetonitrile and 0.1% formic acid in water. For data acquisition, a FAIMS (i.e. field asymmetric ion mobility spectrometry) device was attached and set to a constant voltage of –45 V. The mass spectrometer was operated in data independent acquisition mode with a cycle time of 3 s. MS1 scans were acquired in the Orbitrap in centroid mode at a resolution of 60,000 FWHM (i.e. full width at half-maximum) (*m/z*

200), a scan range from 400 to 1000 m/z, normalized AGC target set to 250% and maximum ion injection time mode set to 50 ms. MS2 scans were acquired in the Orbitrap in centroid mode at a resolution of 120,000 FWHM (*m/z* 200), precursor mass range of 400 to 900, quadrupole isolation window of *m/z* 56 with *m/z* 1 window overlap, normalized automatic gain control target set to 2000% and a maximum injection time of 246 ms. Peptides were fragmented by HCD (i.e. higher-energy collisional dissociation) with collision energy set to 27% and one microscan was acquired for each spectrum.

The acquired raw-files were searched by direct data independent acquisition against the murine UniProt protein database (version February 2022, https://www.uniprot.org/) and commonly observed contaminants by the SpectroNaut software (Biognosys AG, Wagistrasse 21, 8952 Schlieren, Switzerland, version 18.6.231227.55695) using default settings. The search criteria were set as follows: full tryptic specificity was required (cleavage after lysine or arginine residues unless followed by proline), two missed cleavages were allowed, carbamidomethylation (C) was set as fixed modification and oxidation (M) and N-terminal acetylation as a variable modification. The false identification rate was set to 1%. The search results were exported from SpectroNaut and protein abundances were statistically analysed using MSstats (v. 4.4.1, https://www.bioconductor.org/packages/release/bioc/html/MSstats.html). Empirical Bayes moderated *t* tests were applied and the resulting *P* values per protein comparison were adjusted for multiple testing using the Benjamini–Hochberg method (Choi et al., 2014). Genes showing a ≥2.0-fold change in protein expression ($P \leq 0.05$) were analysed by DAVID functional annotation to produce gene clusters (≥2 genes/cluster). Gene Ontology (GO) (http://geneontology.org) terms corresponding to biological process (GOTERM_BP_FAT and KEGG_PATHWAY) were extracted and plotted with the numbers of genes (as a percentage of the total) for each term. GO terms with <2% of the total genes were not plotted unless significantly enriched (Benjamini ≤0.05).

### *In vivo* proprioceptor functional tests using the Beam walk and CatWalk tests

All behavioural tests and analyses were conducted by an experimenter blinded to the mouse genotype. Mice were investigated for 10 weeks, starting from the age of 6 weeks. The first week was used as acclimatization to the experimental device. In the case of the Beam walk test, balance was tested on a 1 m long, 1 cm wide beam suspended on two poles 50 cm above a table top. Food was placed in a house-like goal structure at the end-point to attract the mouse. A video camera

on a tripod was used to record the walk and help the experimenter assign the score for each mouse. Scores were given on a scale of 0–3, with zero representing absence of any pathological phenotype and 3 representing a compromised phenotype (Fig. 1*B* for score assignment details). Beam walk measurements were conducted four times per week and the last two measurements were used for the statistical analysis because these were the ones when mice were the most acquainted to the experimental procedure. The CatWalk XT system (v10.0.408) (Noldus, Leesburg, VA, USA) was used to assess qualitative and quantitative differences between WT, Ex36 and dHT age matched male mice following published experimental protocols (Kryaiku et al., 2016). Fig. 1*C* describes how parameters were assessed. Prior to testing, mice were acclimatized to the dark room and the CatWalk XT instrument over 3 days for 1 h day$^{-1}$ with the illuminated surface turned on. The CatWalk system automatically recorded videos of the mice walking the entire length of the walkway. Experimental sessions typically lasted 5–10 min, and three compliant runs were recorded per mouse. Successful runs were established as a continuous and straight walk, without interruptions or head turning to the sides. Camera gains was set to: 20 dB, detection threshold of 0.1 with red ceiling light of 17.7 V and green walkway of 16.5 V. Experiments were conducted at the same time of the day and a minimum of three successful runs was acquired. The CatWalk system determined the compliance (60%) according to the run's duration and speed variation. Paw positions were verified manually after acquisition. Videos were classified and data were analysed using custom software and the results of three compliant runs per mouse were averaged and plotted.

### X-ray microtomography

Twelve-week-old male mice were killed by i.v. injection of pentobarbital and immersed overnight in PBS containing 4% paraformaldehyde. Whole mice were examined by X-ray microtomography using a SKYSCAN 1275 (Bruker, Kontich, Belgium) equipped with a 100 kV/10 W microfocus X-ray source. All scans were performed with an accelerating voltage of 80 kV and a beam current of 125 μA. To increase the mean energy of the X-ray spectrum, a 0.8 mm Al filter was used. The effective pixel size was set to 20 μm and the exposure time to 0.63 s per rotation angle. In total, 1440 equiangular projections were acquired over the angular range of 360°, resulting in an acquisition time of ~1.5 h per scan. Because of the size of the mouse, five height steps were recorded. The total acquisition time for the entire mouse was therefore around 7.5 h. The projections were reconstructed with a cone beam filtered back projection algorithm using the manufacturer's NRecon (version 1.7.4.6) software

(Bruker). VGStudio MAX 2.1 software (Volume Graphics GmbH, Heidelberg, Germany) was used for visualization and extraction of vertebral co-ordinates. A digital 3-D reconstruction of the spine was performed to examine scoliosis and kyphosis. The Cobb angles were measured with Fiji (https://doi.org/10.1038/nmeth.2019).

For Kyphosis, a permutation statistical test, cubic splines were fit to the 3-D point clouds of each mouse in the dataset (https://zenodo.org/doi/10.5281/zenodo.12721977). These smooth curves were constrained to be flat (zero curvature) at their extremes, corresponding to the points that would lead to the mouse tail and head connections. That was made to focus on the inner part of the annotated spine columns and avoid spurious curvature at the extremes.

The cubic spline fit leads to a parametrization of the type $\gamma = \gamma(t) = (\gamma_x(t), \gamma_y(t), \gamma_z(t))$ for each spine column, where $t$ is a traversal parameter in the interval $[0,1]$ such that $\gamma(0)$ and $\gamma(1)$ are the points at the beginning and end of the parametrized smooth curve, respectively. The curvature of a parametrized curve was $k(t)$, computed at each $\gamma(t)$ value of $t$ using:

$$k(t) = \frac{\|\gamma'(t) \times \gamma''(t)\|}{\|\gamma'(t)\|^3}$$

where $\gamma'$ and $\gamma''$ are the first and second derivate $\gamma$ with respect to t, and $\| \cdot \|$ is the usual Euclidean norm.

The radius of curvature of $\gamma(t)$ at each $t \in [0, 1]$ is given by the reciprocal of the curvature:

$$r(t) = \frac{1}{k(t)}$$

We compare populations of mice via summaries of the curvatures of the smooth curves fitted to their spine column point clouds. The code used for the analysis is available online (https://gitlab.com/ceda-unibas/spine-curvatures)

### Intrafusal muscle fibre isolation and calcium measurements

Six-week-old WT, Ex36 and dHT male mice were killed by pentobarbital overdose according to the procedures approved by the Kantonal Veterinary Authority. EDL muscles were isolated and bathed in Dulbecco's modified Eagle's medium (Sigma). The four distal tendons of the EDL muscles were used to dissect the four proximal ends of the EDL muscles under a SZX12 Stereo Microscope (Olympus, Tokyo, Japan) to obtain four single muscle sections. These were then pinned to the isolation chamber and the extrafusal fibres surrounding the muscle spindles were dissected out mechanically.

The exposed muscle spindles were incubated at 25°C in Tyrode's solution (137 mM NaCl, 5.4 mM KCl, 0.5 mM

$MgCl_2$, 1.8 mM $CaCl_2$, 0.1% glucose and 11.8 mM HEPES, adjusted to pH 7.4 with NaOH) containing 10 μM of the calcium indicator Fura-Red AM (Thermo Fisher Scientific) and 50 μM *N*-benzyl-*p*-toluene sulfonamide (Tocris, Zug, Switzerland). After 50 min of incubation at 25°C, the fibres were rinsed with fresh Tyrode's solution and placed in Tyrode's solution containing 50 μM *N*-benzyl-*p*-toluene sulfonamide. Online intracellular calcium measurements were performed in Fura-Red loaded fibres, using a Eclipse TE2000-E fluorescent microscope (Nikon) equipped with a 20× Plan Apo VC Nikon objective (N.A. = 1.4). Fibres were electrically stimulated by field stimulation with a single pulse of 50 V of 1 ms duration delivered with a Grass S88 Stimulator (Grass Technology, Middleton, WI, USA). Fura-Red was excited at 405 nm (D-Eclipse C1 laser; Nikon) and emission was recorded at 655 nm using a Laser Beamsplitter zt 405 RDC filter (AOI 0°; 655/15 BrightLine HC; AHF analysentechnik Tubingen, Germany). Changes in Fura-Red fluorescence were monitored using an electron multiplier CCD camera (C9100-13; Hamamatsu Photonics, Hamamatsu City, Japan); images were captured by using a subarray of $32 \times 32$ pixels, binning $2 \times 2$ at a frequency of 300 frames $s^{-1}$ (0.3 kHz). The acquired images were analysed using Metamorph 5.7.4 software (Molecular Devices, San Jose, CA, USA); the results are reported as $\Delta F/F_0 = (F_{max} - F_{rest})/(F_{rest})$. Kinetic parameters were analysed as previously described (Mosca et al., 2013). Calcium measurements were carried out on one intrafusal muscle fibre, isolated from one EDL per mouse, on a total of four mice per genotype.

### Live muscle spindle imaging

Isolated muscle spindles were incubated with DAPI for 30 min at 25°C in Tyrode's solution, rinsed and then observed under an A1plus confocal microscope (Nikon) equipped with Nikon Plan Apo, 40× oil objective (N.A. = 1.4) and with a Sapphire laser (405 nm) (Coherent, Saxonburg, PA, USA) controlled by NIS-Elements Confocal software (version 4.6) (Nikon). Images were analysed using Fiji software (version 3.1.3) (NIH).

### Statistical analysis

Data are plotted as the mean ± SD. Statistical analysis was performed using the Origin (Pro), version 2019 (OriginLab Corp., Northampton, MA, USA) using the Mann–Whitney test unless otherwise specified. $P < 0.05$ was considered statistically significant.

## Results

### RyR1 is expressed in intrafusal fibres

To demonstrate the presence of RyR1 in muscle spindles, we collected samples of intrafusal muscle fibres and extrafusal muscle fibres from WT mice by laser capture microdissection (LCM) (Fig. 1*A*) and performed quantitative proteomic analysis of the LCM samples. The muscle spindles were first identified in the cryosection, and then microdissection was carried out at the polar region of the spindles where the thickness of the gap separating intrafusal fibres from extrafusal fibres is small. As experimental negative controls, we also collected liver and kidney samples from WT mice and analysed them using mass spectrometry. Figure 2*A* shows the principal component analysis (PCA) analysis of the proteomic data supporting the appropriateness of our experimental approach. In particular, the proteome of non-muscle tissues such as liver (Fig. 2*A*, light brown samples) and kidney (Fig. 2*A*, red samples) is very different from that of skeletal muscles (Fig. 2A, green and violet samples) and the expression level of RyR1, if any, in liver and kidney is close to zero (Fig. 2*B*), confirming the specificity of the quantitative mass spectrometry analysis. Importantly, extrafusal muscle fibres (Fig. 2*A*, green samples) can be differentiated from intrafusal muscle fibres (Fig. 2*A* violet samples) by PCA analysis. Of note, extrafusal fibres contain 2.05-fold ($P = 0.0028$) more RyR1 protein compared to intrafusal fibres (Fig. 2*B*). On the basis of previously published quantitative mass spectrometry analysis of extrafusal fibres using spiked-in labelled peptides from RyR1 (Eckhardt et al., 2023), the calculated average of RyR1 protein content in intrafusal fibres of WT mice is 0.628 μmol $kg^{-1}$ wet weight and the calculated RyR1 tetrameric complex is 0.152 μmol $kg^{-1}$ wet weight.

Having unequivocally shown that intrafusal fibres express an assessable amount of RyR1, we next investigated its subcellular localization by confocal immunohistochemistry. Longitudinal section of EDL and soleus muscles from Thy-1- EYFP transgenic mice were stained with anti-skeletal muscle RyR1 antibodies (Ab) and DAPI. Thy-1 marks all sensory neuron projections including group Ia and group II sensory afferent endings in the equatorial region of intrafusal muscle fibres (Chelser et al., 2016). This area of intrafusal nuclear bag fibres is enriched in nuclei as indicated by DAPI staining (Fig. 2*C*, left). The equatorial region extends into the polar region. The RyR1 Ab preferentially stains the polar regions of the intrafusal fibres (white arrows) but not the central nuclear bag region (empty arrows). The density profile of the RyR1 Ab staining shows the doubled row pattern typically observed in proteins located in the junctional SR of extrafusal fibres (DiFranco et al., 2011) (Fig. 2*C*, right). No apparent difference in DAPI or

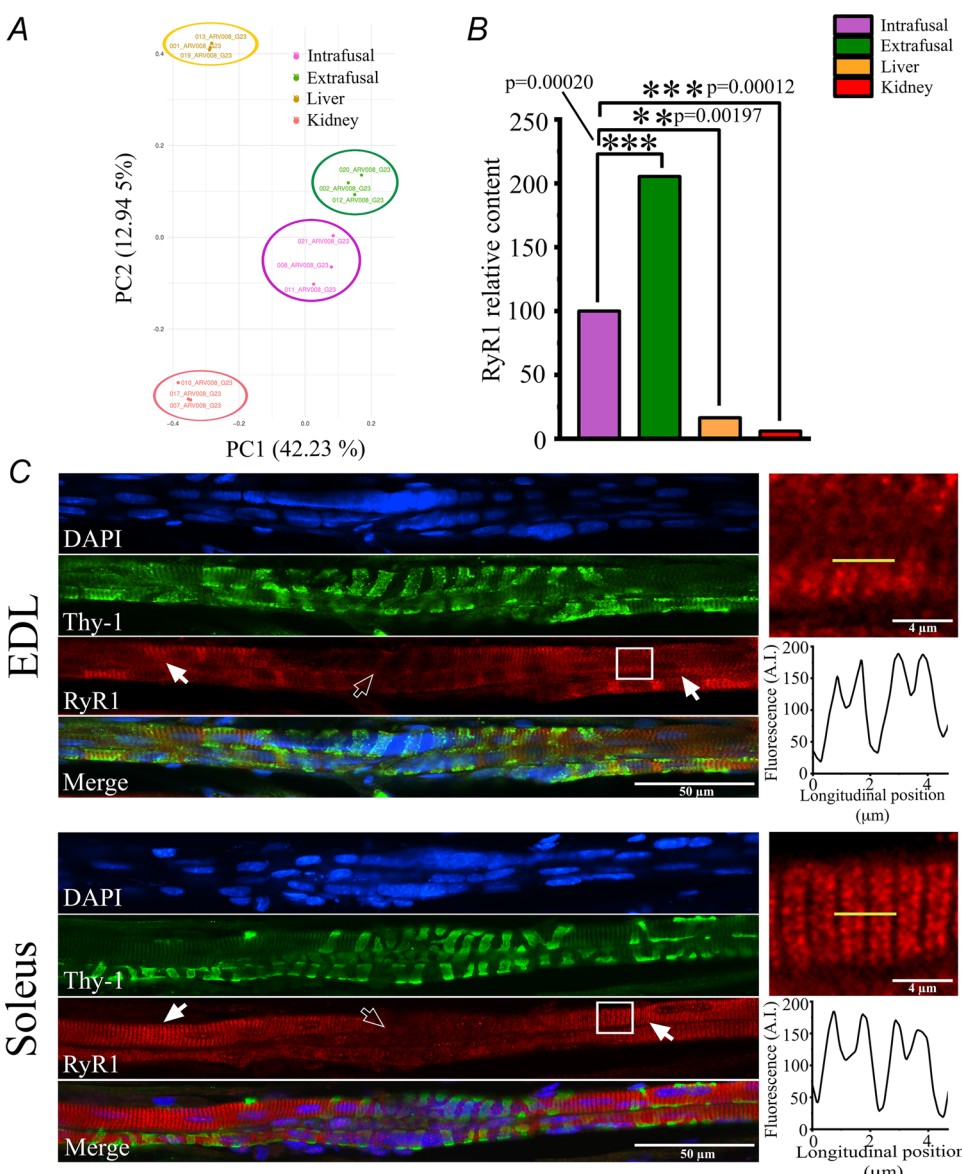

**Figure 2. Intrafusal muscle fibres express RyR1 protein and have a protein composition that is distinct from that of extrafusal muscle fibres**

*A*, PCA analysis of the proteome of intrafusal muscles (violet), extrafusal muscles (green), liver (brown) and kidney (red) from WT mice. Each symbol represents the analysis of proteins from WT mice ($n = 3$ mice). Intrafusal and extrafusal muscles were isolated form soleus muscles. *B*, bar charts comparing relative RyR1 content in intrafusal fibres (soleus + EDL, set as 100%), extrafusal soleus fibres, kidney and liver. RyR1 protein levels were obtained by mass spectrometry analysis on microdissected samples. Samples isolated from the intrafusal fibres, were selectively collected from the polar region of the fibre. Acquired reporter ion intensities in the experiments were employed for automated quantification and statistically analysed using a modified version of our in-house developed SafeQuant R script (v2.3), the calculated *P* values were corrected for multiple testing using the Benjamini−Hochberg method. Extrafusal, ****P* = 0.000204; liver, ***P* = 0.00197; kidney, ****P* = 0.000124. *C*, subcellular localization of RyR1 protein in muscle spindle fibres from 12 week WT mice. Confocal images were acquired along the longitudinal axis of muscle spindle fibres from EDL and soleus muscles. Images show the general structure of the muscle spindle, containing a polar region (white arrows) where RyR1 positive immunofluorescence is highest and the equatorial region (empty arrows) with a high content of nuclei (nuclear bag). Fibres were fixed and processed as described in the Methods, and stained with DAPI (blue), Thy-1 (green) and RyR1 (red). Images were acquired using a Nikon AxR confocal microscope equipped with a $40\times$ Plan Apo VC Nikon objective (N.A. = 0.95). Scale bars = 50 µm. The two insets show a higher magnification of the boxed region in EDL and soleus RyR1 staining, where the double labelling pattern characteristic of RyR1 distribution is clearly visible. White magnification bar = 4 µm. The yellow bar shows the longitudinal position of the fluorescence labelled RyR1. The lower panels of the two insets show the longitudinal spatial fluorescence profiles over the transverse direction within the line.

RyR1 staining was observed between the muscle spindles of EDL and soleus muscles (Fig. 2*C*).

### Proteomic analysis reveals significant differences in the protein composition of intrafusal *vs.* extrafusal muscles

The protein content of intrafusal and extrafusal fibres was analysed under stringent conditions by removing proteins having ≤2 peptides, exhibiting a fold change ≥0.20 ($-0.321 \leq \log_2FC \leq 0.263$) and $q \leq 0.05$. By filtering the mass spectrometry results using these stringent parameters, our results show that intrafusal muscle fibres have a significantly different protein composition compared to extrafusal fibres, showing 542 down-regulated and 1591 up-regulated proteins (Fig. 3*A*). GO pathway analysis of the up-regulated

and down-regulated proteins in intrafusal *vs.* extrafusal muscle fibres identified proteins preferentially clustering to biological processes, molecular function and local network (Fig. 3*B* and 3*C*). Within the biological process category, the pathways showing the largest enrichment score of up-regulated genes belong to the filopodium assembly, astrocyte development and xenobiotic transport groups. Within the molecular function category, the pathways showing the largest enrichment score of up-regulated genes belong to the xenobiotic membrane transport activity, fibronectin binding and extracellular matrix binding groups. Within the local network cluster category, the pathways showing the largest enrichment score of the up-regulated genes belong to the neuregulin binding, integrin alphav complex, aggrecan/versican and amino acid transport groups (Fig. 3*B*). Regarding the down-regulated pathways, within the biological process category, the pathways showing the largest

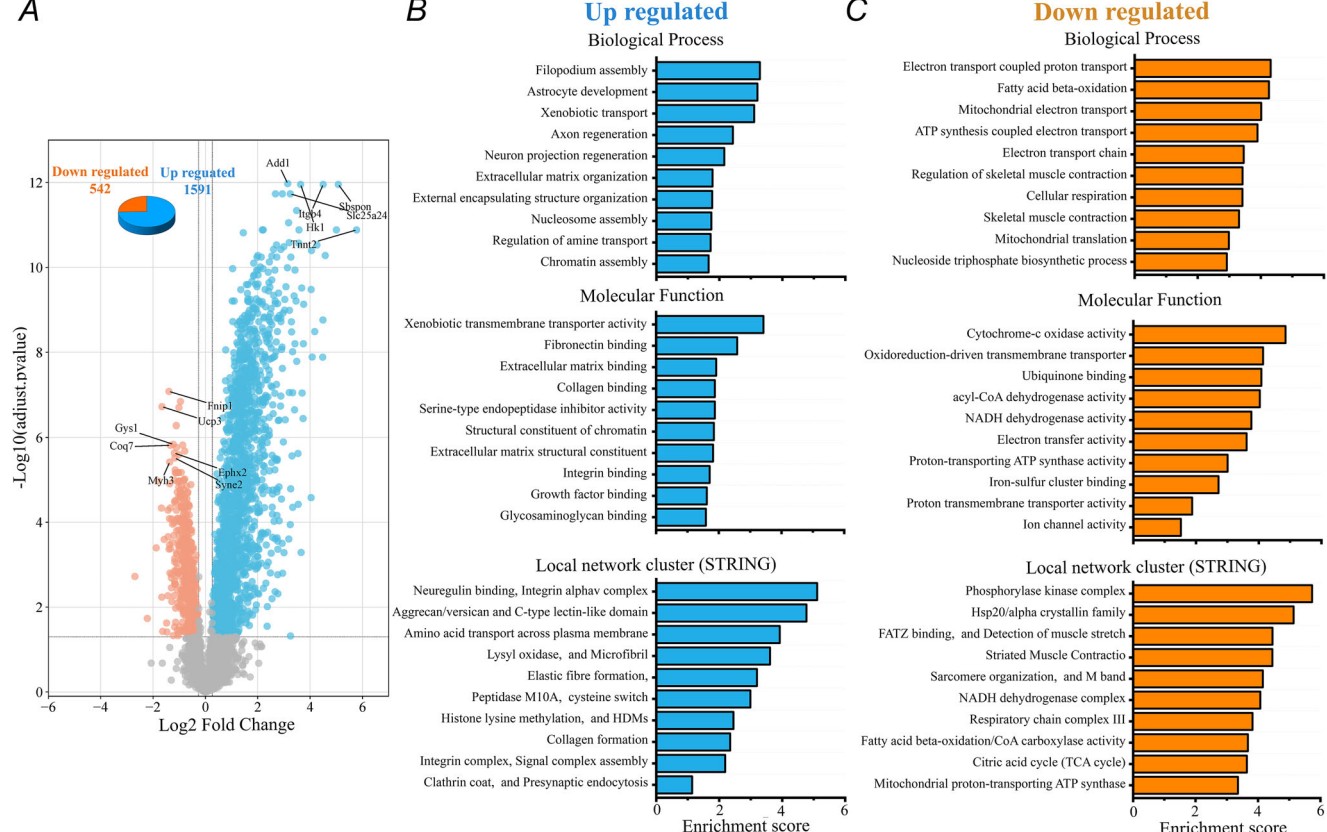

**Figure 3. Comparison of the protein composition of intrafusal *vs.* extrafusal muscle fibres from WT mice**
*A*, volcano plot analysis comparing protein content of intrafusal and extrafusal muscle fibres from soleus muscles of WT mice. The data was filtered so that only proteins quantified based on ≥2 peptides exhibiting fold change ≥0.20 ($-0.321 \leq \log_2FC \leq 0.263$) and $q \leq 0.05$ were considered as differentially expressed. More than 500 proteins were down-regulated in intrafusal *vs.* extrafusal muscle fibres, whereas more than 1500 were enriched in intrafusal *vs.* extrafusal muscle fibres. *B*, proteins showing a significant enrichment in intrafusal *vs.* extrafusal muscles were analysed by GO pathway analysis. *C*, proteins showing a significant depletion in intrafusal *vs.* extrafusal muscles were analysed by GO pathway analysis. In both (*B*) and (*C*), the enrichment scores were calculated based on $\log_2$ FC for protein quantification based on ≥2 peptides. Data are plotted as Enrichment score *vs.* significantly altered GO pathways. Muscles were collected from five WT mice.

down-regulated enrichment score belong to the electron transport coupled proton transport, fatty acid ß-oxidation and mitochondrial electron transport. Within the molecular function category, the pathways showing the largest down-regulated enrichment score belong to cytochrome *c* oxidase activity, oxidoreduction-driven transmembrane transport and ubiquinone binding. Within the local network cluster category, the pathways showing the largest down-regulated enrichment score belong to phosphorylase kinase complex, Hsp20/$\alpha$ crystallin family and FATZ binding and detection of muscle stretch (Fig. 3*C*). Table 2 shows the 50 top proteins showing the greatest fold change in intrafusal *vs.* extrafusal muscles from WT mice. Cardiac troponin T (TnnT2), somatomedin B and thrombospondin type 1 domain containing protein (Sbspon), dipeptidyl peptidase 4 (Dpp4), monooxygenase DBH like 1 (Moxd1), integrin ß4 subunit (Itgb4) and collagen type XXVIII alpha 1 chain (Col28a1) were the most enriched in intrafusal muscle fibres and nicotinamide nucleotide transhydrogenase (Nnt), myosin-binding protein C2 family (Mybcp2), aquaporin 7 (Aqp7), phosphorylase b kinase gamma catalytic chain (Phkg1) and trio rho guanine nucleotide exchange factor (Trio) were the least enriched in intrafusal muscles from WT mice. We also looked for markers of neuronal components and/or nerve endings within the muscle spindle proteome. In particular, we searched for those neuronal markers identified by Bornstein et al. (2023) and, in agreement with their analysis, we found that the following proteins were enriched in muscle spindles compared to extrafusal fibres: $\alpha$3 subunit of the Na$^+$/K$^+$ ATPase (*Atp1a3*, present in both afferent and efferent neurons and in the membranes of spiral ganglion somata; fold change = 10.479), dihydropyrimidinase like 3 (*Dpysl3*, present in both afferent and efferent nerves, especially in motor neurons; fold change = 4.50), heat shock 70 kDa protein 12A (*Hspa12a*, present in both afferent and efferent neurons as well as other cellular compartments; fold change = 11.194), vesicle amine transport-1-like (*Vat1*, primarily located in efferent nerves, specifically in the nerve terminals of cholinergic neurons involved in the release of acetylcholine; fold change = 3.119), peptidyl arginine deiminase 2 (*Plp1*, the major myelin protein of the CNS and therefore is present in both afferent and efferent neurons; fold change = 11.865) and class 3 ß-tubulin (*Tubb3*, neither afferent nor efferent, but rather playing a structural and functional role in nerve cells; fold change = 4.594). In addition, comparison of the present results with transcriptional and proteomic data (Bornstein et al., 2023; Kim et al., 2020) of muscle spindles *vs.* extrafusal muscles confirm that muscle spindles are also enriched in a number of other proteins including, acetylcholinesterase

(fold change = 1.676), collagen 3$\alpha$1 and 6$\alpha$1 (fold change = 2.06 and 2.299 respectively), $\alpha$ subunit of cardiac myosin heavy chain (fold change = 7.029), immunoglobulin-like and fibronectin type III domain containing 1 (fold change = 4.228), high mobility group protein B1 (fold change = 2.535), versican (fold change = 5.291), elastin (fold change = 3.610). fibulin-3 (fibrillin like protein, fold change 6.485), periostin (fold change = 18.099), myelin proteolipid protein 1 (fold change = 6.570) and myosin light chain 4 (fold change = 11.7) (Table 3). Furthermore, Thy1, which we used as a marker to identify muscle spindles in immuno-histochemical confocal experiments, was enriched by 6.624-fold.

The mass spectrometry data have been deposited in the ProteomeXchange Consortium via the ProteoSAFe repository (http://massive.ucsd.edu/ProteoSAFe) under access number PXD054222 (https://massive.ucsd.edu/ProteoSAFe/private-dataset.jsp?task = 09ff33af463e430f8e40ce18bf5cf1d2).

In summary, proteomic analysis demonstrates that RyR1 is expressed in intrafusal muscle fibres, along with a large set of other proteins, leading us to next investigate whether *Ryr1* mutations affect muscle spindles in a murine model for a RyR1 linked recessive congenital myopathy. For this study, we used three mouse models that we have developed to investigate the phenotype caused by *Ryr1* mutations, namely, RyR1p.F4976L (HO; Benucci et al., 2024), RyR1 Q1970fx12 (Ex36; Elbaz et al., 2019b) and RyR1p.Q1970fsX16+p.A4329D (dHT; Elbaz et al., 2019a). The latter mouse model carries mutations that are syngeneic to those identified in a severely affected child who died in infancy (patient 28, Zhou et al., 2007; patient 56, Klein et al., 2012). Figure 4 summarizes the major phenotypic features of extrafusal muscle fibres in each mouse model. Briefly, dHT mice display a severe phenotype including a 65% reduction of RyR1 protein content, which leads to a 60% reduction of specific twitch force (Elbaz et al., 2019a). Ex36 mice carry a frameshift mutation leading to a downstream premature stop codon, causing non-sense mediated RNA decay of the mutant allele and resulting in a ∼60% and 40% decrease of RyR1 content in EDL and soleus muscles, respectively (Elbaz et al., 2019b). The decrease in RyR1 protein content in extrafusal muscles was paralleled by an average 19% and 28% decrease of electrically evoked force generated in isolated EDL and soleus muscles, respectively. HO mice, a transgenic mouse line carrying the homozygous RyR1 mutation p.F4976L syngeneic to that carried by a severely affected child (Benucci et al., 2024), exhibit the smallest reduction of RyR1 content, with an average 36% and 28% reduction of electrically evoked force development in EDL and soleus muscles, respectively.

**Table 2. Top 50 proteins showing the greatest fold change in content in intrafusal *vs.* extrafusal muscle fibres in WT mice**

| Top 50 Up-regulated | | | Top 50 Down-regulated | | |
| --- | --- | --- | --- | --- | --- |
| Gene name | Fold change | *q* value | Gene name | Fold change | *q* value |
| Tnnt2 | 54.785 | 0.019921 | Nnt | 0.1547 | $1.11 \times 10^{-12}$ |
| Sbspon | 33.795 | 0.001899 | Mybpc2 | 0.2149 | $1.31 \times 10^{-11}$ |
| Dpp4 | 32.134 | 0.018494 | Aqp7 | 0.2719 | $5.25 \times 10^{-11}$ |
| Moxd1 | 23.913 | 0.000402 | Phkg1 | 0.2869 | $1.11 \times 10^{-12}$ |
| Itgb4 | 22.624 | $1.08 \times 10^{-5}$ | Trio | 0.3120 | $1.74 \times 10^{-9}$ |
| Col28a1 | 22.486 | $4.63 \times 10^{-5}$ | Abra | 0.3158 | $1.31E \times 10^{-8}$ |
| Sgce | 22.362 | 0.005811 | Ucp3 | 0.3169 | $2.97 \times 10^{-11}$ |
| Atp1b3 | 19.165 | $1.92 \times 10^{-7}$ | Tpm1 | 0.3225 | $3.26 \times 10^{-9}$ |
| Postn | 18.099 | 0.037256 | Tmod4 | 0.3355 | $1.30 \times 10^{-8}$ |
| Fbln2 | 16.913 | 0.000254 | Myl2 | 0.3582 | $4.01 \times 10^{-11}$ |
| Slc43a1 | 16.609 | 0.023672 | Mtx2 | 0.3614 | $3.32 \times 10^{-10}$ |
| Col18a1 | 16.458 | 0.038686 | Pak1 | 0.3706 | $2.63 \times 10^{-5}$ |
| Cend1 | 16.439 | 0.005393 | Cox7a1 | 0.3713 | $7.39 \times 10^{-10}$ |
| Akap12 | 13.495 | $5.18 \times 10^{-5}$ | Cox7b | 0.3727 | $6.31 \times 10^{-9}$ |
| Fbln1 | 13.388 | 0.000219 | Fnip1 | 0.3809 | $3.66 \times 10^{-7}$ |
| Olfml3 | 13.298 | $8.39 \times 10^{-8}$ | Fabp3 | 0.3823 | $9.15 \times 10^{-8}$ |
| Slc4a4 | 12.962 | 0.00043 | Fsd2 | 0.3863 | $3.06 \times 10^{-10}$ |
| Kif21a | 12.817 | $8.23 \times 10^{-5}$ | Fbp2 | 0.3875 | 0.00051582 |
| Thsd4 | 12.621 | 0.000338 | Myh3 | 0.3884 | $1.11 \times 10^{-12}$ |
| Hk1 | 12.454 | $3.82 \times 10^{-6}$ | Retreg1 | 0.3892 | $1.31 \times 10^{-11}$ |
| Itga6 | 11.919 | $1.24 \times 10^{-5}$ | Pxmp2 | 0.3916 | $2.69 \times 10^{-11}$ |
| Padi2 | 11.865 | 0.001573 | Apobec2 | 0.3952 | $3.78 \times 10^{-5}$ |
| Myl4 | 11.700 | $4.59 \times 10^{-5}$ | Coq7 | 0.3956 | $2.40 \times 10^{-6}$ |
| Hapln1 | 11.613 | $1.54 \times 10^{-6}$ | Ddo | 0.3975 | $1.06 \times 10^{-12}$ |
| Add3 | 11.265 | 0.000416 | Prr33 | 0.4062 | $1.64 \times 10^{-8}$ |
| Igfbp6 | 11.232 | 0.002048 | Hspb7 | 0.4067 | $4.58 \times 10^{-12}$ |
| Hspa12a | 11.194 | 0.003513 | Myoz2 | 0.4083 | $6.21 \times 10^{-10}$ |
| Lpar1 | 11.041 | 0.001682 | Hspb6 | 0.4102 | $1.16E-10$ |
| Abcb1a | 10.720 | 0.007356 | Etfrf1 | 0.4103 | $6.31E-09$ |
| Htra1 | 10.715 | $8.35 \times 10^{-5}$ | Ndufb8 | 0.4152 | $2.28E-07$ |
| Fzd2 | 10.606 | 0.000301 | Myh8 | 0.4198 | $4.72E-10$ |
| Atp1a3 | 10.479 | 0.000111 | Casq1 | 0.4270 | $4.45E-05$ |
| Enpp1 | 10.098 | 0.046816 | Gys1 | 0.4274 | $8.99E-08$ |
| Igsf8 | 10.082 | $1.46 \times 10^{-6}$ | Flnc | 0.4327 | $9.74E-09$ |
| Xpnpep2 | 10.070 | 0.000205 | Phka1 | 0.4372 | $2.73 \times 10^{-6}$ |
| Rab34 | 9.911 | 0.000105 | Mylpf | 0.4387 | $3.32 \times 10^{-9}$ |
| Isyna1 | 9.818 | 0.005342 | Ephx2 | 0.4398 | $1.92 \times 10^{-9}$ |
| Col12a1 | 9.758 | $2.34 \times 10^{-6}$ | Des | 0.4403 | $5.73 \times 10^{-11}$ |
| Sfxn3 | 9.610 | $1.02 \times 10^{-5}$ | Mief2 | 0.4413 | $1.63 \times 10^{-5}$ |
| Atp6v1g2 | 9.576 | 0.002403 | Acsl1 | 0.4417 | 0.04790694 |
| S100a6 | 9.520 | $3.25 \times 10^{-5}$ | Ubr3 | 0.4424 | $1.85 \times 10^{-12}$ |
| Slc25a24 | 9.477 | 0.02924 | Scn4a | 0.4432 | 0.00011362 |
| Clec3a | 9.465 | 0.00072 | Perm1 | 0.4438 | $1.03 \times 10^{-9}$ |
| IGHG3 | 9.263 | 0.000854 | Acad11 | 0.4441 | $8.91 \times 10^{-8}$ |
| Igfbp5 | 9.227 | 0.00139 | Aqp1 | 0.4448 | $2.61 \times 10^{-11}$ |
| Sntb1 | 9.205 | 0.002906 | N4bp1 | 0.4454 | 0.00024887 |
| Smarcd3 | 9.110 | 0.002818 | Syne2 | 0.4463 | $8.87 \times 10^{-12}$ |
| Epb41l2 | 9.057 | $3.06 \times 10^{-6}$ | Acadvl | 0.4490 | $1.06 \times 10^{-12}$ |
| Add1 | 8.841 | $5.84 \times 10^{-6}$ | Higd1a | 0.4502 | $1.02 \times 10^{-10}$ |
| Itih5 | 8.781 | 0.001438 | Myot | 0.4503 | $9.05 \times 10^{-5}$ |

**Table 3. Enriched proteins (values are intrafusal *vs.* extrafusal muscle fibres) identified in the present study as well as previously reported (Bornstein et al., 2023; Kim et al., 2020)**

| Gene name | Fold change | *q* value | Location | Reference |
| --- | --- | --- | --- | --- |
| *Synm* | 0.523 | 0.0049 | Chain2 | Kim et al. (2020) |
| *Mylk2* | 0.545 | 0.00079 | Chain2+Bag+spdNMJ | Kim et al. (2020) |
| *Tnnt3* | 0.583 | 0.011 | Chain1, Chain2+Bag+spdNMJ+spdMTJ | Kim et al. (2020) |
| *Myom1* | 0.617 | 0.00037 | Chain1, Chain2+Bag+spdNMJ | Kim et al. (2020) |
| *Ache* | 1.676 | 0.017 | Bag+spdNMJ | Kim et al. (2020) |
| *Col3a1* | 2.064 | 0.001 | Chain+Bag+spdMTJ | Kim et al. (2020) |
| *Col6a1* | 2.299 | $1.11 \times 10^{-6}$ | spdMTJ | Kim et al. (2020) |
| *Hmgb1* | 2.537 | $8.71 \times 10^{-8}$ | Chain1, Chain2+Bag+spdNMJ | Kim et al. (2020) |
| *Myl6b* | 2.320 | 0.00027 | Intrafusal fibre | Bornstein et al. (2023) |
| *Vat1l* | 3.119 | 0.012 | Neuron | Bornstein et al. (2023) |
| *Eln* | 3.610 | $1.0 \times 10^{-8}$ | Capsule | Bornstein et al. (2023) |
| *Igfn1* | 4.228 | 0.00026 | Chain1, Chain2+Bag+spdNMJ+spdMTJ | Kim et al. (2020) |
| *Dpysl3* | 4.507 | $6.44 \times 10^{-10}$ | Neuron | Bornstein et al. (2023) |
| *Tubb3* | 4.594 | $1.53 \times 10^{-7}$ | Neuron | Bornstein et al. (2023) |
| *Vcan* | 5.291 | 0.00087 | Capsule | Bornstein et al. (2023) |
| *Efemp1* | 6.485 | $9.74 \times 10^{-9}$ | Capsule | Bornstein et al. (2023) |
| *Plp1* | 6.570 | $5.74 \times 10^{-8}$ | Neuron | Bornstein et al. (2023) |
| *Thy1* | 6.624 | $1.58 \times 10^{-9}$ | Neuron projections including the group Ia and group II sensory afferent endings in the equatorial region of intrafusal muscle fibres | Kröger & Watkins (2021) |
| *Myh6* | 7.029 | $1.47 \times 10^{-5}$ | Bag | Bornstein et al. (2023) |
| *ATP1a3* | 10.482 | $4.72 \times 10^{-10}$ | Neuron | Bornstein et al. (2023) |
| *Hspa12a* | 11.194 | $4.58 \times 10^{-12}$ | Neuron | Bornstein et al. (2023) |
| *Myl4* | 11.700 | $3.78 \times 10^{-5}$ | Intrafusal fibre | Bornstein et al. (2023) |
| *Padi2* | 11.876 | $2.69 \times 10^{-11}$ | Intrafusal fibre | Bornstein et al. (2023) |
| *Postn* | 18.099 | $3.2 \times 10^{-9}$ | Capsule | (Bornstein et al. (2023) |

### Expression of compound heterozygous mutant RyR1 in intrafusal fibres affects the gross histological appearance of muscle spindles

To investigate whether the presence of *Ryr1* mutations affects the gross morphological appearance of intrafusal muscle fibres, we performed serial sections of soleus muscles isolated from 12-week-old WT and transgenic mutant mice. Figure 5*A* shows a schematic representation of the experimental approach and Fig. 5*B* shows representative H&E-stained sections of isolated soleus muscles from WT, dHT, Ex36 and HO mice. Once a muscle spindle was clearly identified in the section (Fig. 5*B*_a, empty arrowheads) subsequent sections were made at a distance of 50 μm and then every 10 μm until the spindle was no longer visible (Fig. 5*A* and *B*). The spindles present in the muscles of all mouse models are embedded within, and surrounded by, extrafusal fibres (white arrowheads); interestingly, intrafusal muscle fibres of dHT mice are surrounded by a wider gap between the muscle spindle and the extrafusal fibres compared to those of WT. The maximal diameter (equatorial region)

of the muscle spindle was identified by scrolling through the individual stack images (Gartych et al., 2021) and was calculated as described in the Methods. The mean (± SD) maximal diameter of muscle spindles of dHT mice is $71.2 \pm 5.8$ μm ($n = 15$ spindles from $n = 5$ mice) and larger than that of spindles of WT mice ($60.6 \pm 7.2$ μm) ($n = 8$ spindles from $n = 3$ mice) ($P = 0.0197$) (Fig. 5C). However, the number of spindles per muscle and their location within the muscle belly, did not differ between WT and dHT mice. The mean ± SD number of muscle spindles in the belly of soleus muscles from WT and dHT mice was $3.5 \pm 0.4$ ($n = 6$) and $3.6 \pm 1.3$ ($n = 11$), respectively ($P = 0.120$). In muscle spindles from WT and dHT mice, the number of nuclear bag and chain fibres was $1.90 \pm 0.29$ and $1.83 \pm 0.38$ (mean ± SD, $n = 18$, $P = 0.491$) and $3.72 \pm 0.88$ and $3.88 \pm 0.83$ (mean ± SD, $n = 22$, $P = 0.676$).

To determine whether the altered histological appearance of the muscle spindles is a consequence of the RyR1p.Q1970fsX16+p.A4329D compound heterozygous mutations and/or occurs in the presence of other RyR1 mutations, we examined H&E stained muscles from transgenic Exon 36 mice (which were used to generate the dHT mice) and HO mice. The monoallelic expression of RyR1 in Exon 36 mice did not result in any gross alterations in spindle morphology or diameters [the mean ± SD spindle diameters of Ex36 and WT mice were $63.2 \pm 9.9$ ($n = 12$) vs. $60.6 \pm 7.6$ ($n = 10$) spindles from $n = 3$ mice, respectively, $P = 0.776$]. Furthermore, the morphology and diameter of the muscle spindle equatorial region in muscles from mice carrying the homozygous RyR1 p.F4976L mutation was not different compared to that of their WT littermates [the mean ± SD spindle diameters of HO and WT mice were $63.4 \pm 7.2$ ($n = 8$) vs. $60.6 \pm 3.6$ ($n = 10$) spindles from $n = 4$ and $n = 3$ mice, respectively, $P = 0.505$].

In conclusion, these results show that in mice carrying the compound heterozygous RyR1p.Q1970fsX16+p.A4329D mutations, but not in those carrying the heterozygous Exon 36 mutation, nor the homozygous RyR1 p.F4976L mutation, the gross morphological appearance of the spindles is altered. We next performed behavioural tests aimed at verifying whether the morphological alterations of muscle spindles observed in the compound heterozygous RyR1p.Q1970fsX16+p.A4329D mice are paralleled by changes in proprioceptive properties such as gait and interlimb co-ordination.

## Gait and interlimb co-ordination properties are compromised in dHT mice

We investigated whether the presence of the RyR1p.Q1970fsX16+p.A4329D compound heterozygous mutations affects balance or causes changes of the gait, by examining the capacity of age and sex-matched WT and dHT littermates to perform the Beam walk and the Catwalk, comprising two behavioural tests that have been shown to be altered in a variety of transgenic mouse models including models with impaired muscle spindle function (Florez-Paz et al., 2016; Taylor et al., 2001). For the Beam walk test, mice were scored for their ability to walk across a 1 m long, 1 cm wide beam suspended on two poles 50 cm above a table top without losing their balance and for their capacity to precisely place their hindlimbs, as described in the Methods and in Fig. 1B (left). All tests were scored by genotype blinded experimenters. In the Supporting information, Videos S1 and S2, as well as Fig. 6A (left), show that dHT mice ($n = 13$) slip when walking across the beam, where they have a significantly higher Beam walk score ($P = 0.0014$) than their WT ($n = 8$) littermates. No significant differences

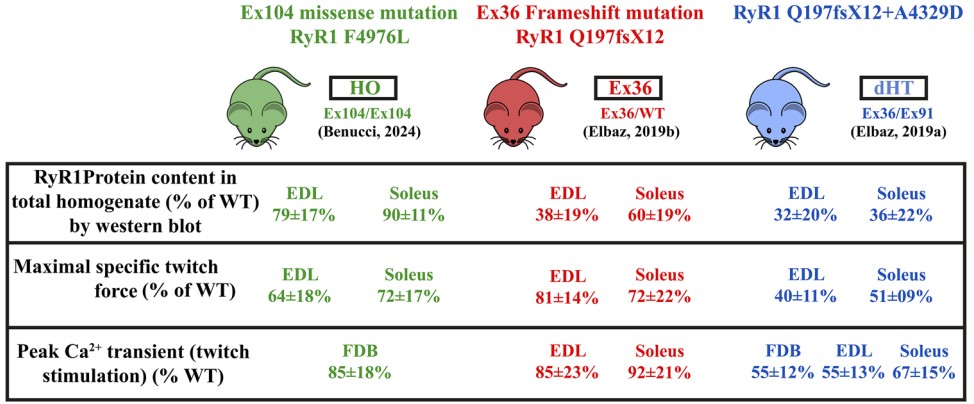

**RyR1 Mutant mouse models of congenital myopathies used in the present investigation**

| | Ex104 missense mutation RyR1 F4976L | | Ex36 Frameshift mutation RyR1 Q197fsX12 | | RyR1 Q197fsX12+A4329D | | |
|---|---|---|---|---|---|---|---|
| | **HO** Ex104/Ex104 (Benucci, 2024) | | **Ex36** Ex36/WT (Elbaz, 2019b) | | **dHT** Ex36/Ex91 (Elbaz, 2019a) | | |
| **RyR1 Protein content in total homogenate (% of WT) by western blot** | EDL 79±17% | Soleus 90±11% | EDL 38±19% | Soleus 60±19% | EDL 32±20% | Soleus 36±22% | |
| **Maximal specific twitch force (% of WT)** | EDL 64±18% | Soleus 72±17% | EDL 81±14% | Soleus 72±22% | EDL 40±11% | Soleus 51±09% | |
| **Peak Ca²⁺ transient (twitch stimulation) (% WT)** | FDB 85±18% | | EDL 85±23% | Soleus 92±21% | FDB 55±12% | EDL 55±13% | Soleus 67±15% |

**Figure 4. Phenotypic characteristics**
Phenotypic characteristics of the transgenic mouse models used in the present investigation.

were observed in the Beam walk score between WT and Ex 36 littermates ($P = 0.18$) (Fig. 6*A*) and WT and HO littermates ($0.067 \pm 0.133$ and $0.083 \pm 0.108$ for WT and HO, respectively; $P = 0.45$). Because all values obtained on HO mice were similar to those of WT littermates, we did not evaluate further HO mice.

The Catwalk is an automated gait analysis system allowing the observer independent and simultaneous quantification of multiple gait parameters (for an explanation of the assessed parameters, see Fig. 1*C*), as well as temporal aspects of interlimb co-ordination (Florez-Paz et al., 2016; Pitzer et al., 2021). Because of

the smaller size and weight of dHT mice, data were normalized to body weight to avoid experimental artefacts. Some CatWalk parameters did not differ between WT and dHT mice, including kinetic parameters and velocity; however, parameters reflecting interlimb

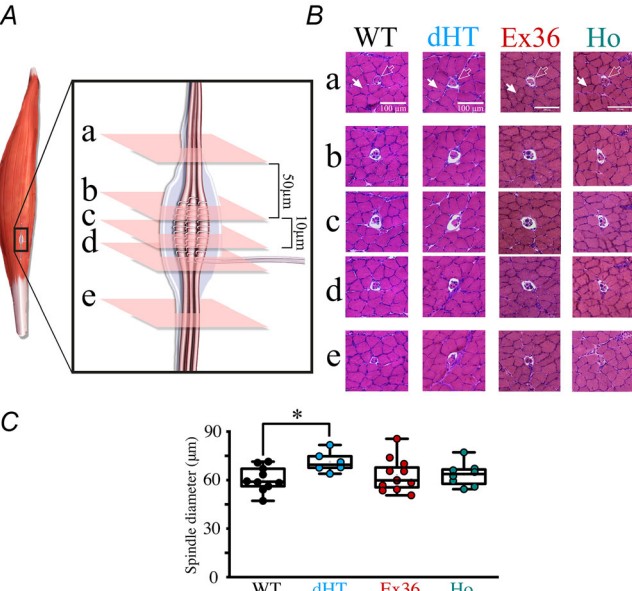

**Figure 5. dHT mice exhibit gross alterations of muscle spindle morphology**

*A*, schematic representation of a skeletal muscle showing the location of a muscle spindle. Skeletal muscles contain several longitudinally oriented muscle spindles ubiquitously distributed in the interior of the muscle belly. They are composed of intrafusal fibres (longitudinally oriented pink fibres) surrounded by a capsule (represented as a light blue matrix). Transverse planes ('a', 'b', 'c', 'd' and 'e') were made through the muscle spindles at five different levels, 'a' and 'e' are located in the polar regions, 'b', 'c' and 'd' are located in the central region. *B*, H&E staining of soleus muscles sections containing intrafusal and extrafusal fibres from WT, dHT, Ex36 and HO mice. Transversal muscle sections (10 µm) were made until the polar region was identified. Extrafusal fibres are indicated by white arrows and intrafusal fibres are indicated by empty arrows. White bar in 'a' = 100 µm. Images were acquired with an Eclipse Ti2 Nikon widefield microscope. *C*, whisker plots of muscle spindle diameter. The diameter was calculated in the equatorial section of the muscle spindle represented as region 'c' in (*A*) and (*B*). The spindle diameter (distance of parallel tangents at opposing borders of the fibre was calculated using Fiji. Each symbol represents results obtained from a single spindle fibre (10 spindles from $n = 3$ WT mice; six spindles from dHT $n = 3$ mice, 12 spindles from $n = 3$ Ex36 mice, eight spindles from $n = 4$ HO mice). *$P = 0.0197$ (Mann–Whitney test).

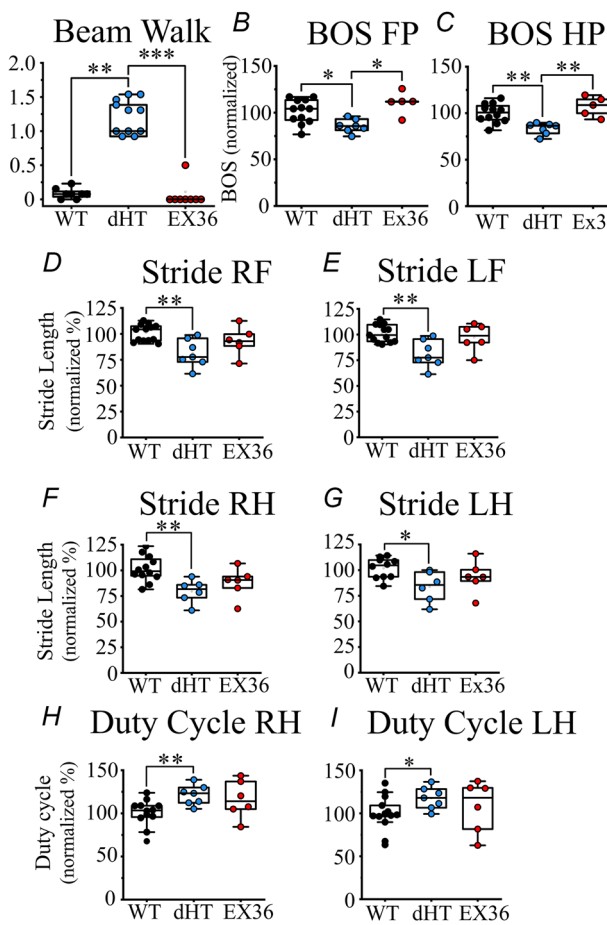

**Figure 6. dHT mice but not Ex36 mice, show proprioceptor function alterations**

*A*, the Beam walk score test of dHT mice is significantly higher than that of WT and Ex36 littermates. Each symbol represents results obtained from a single mouse: WT, $n = 8$; dHT, $n = 12$; Ex36, $n = 9$. WT *vs*. dHT, **$P = 0.0014$; dHT *vs*. Ex36, *$P = 0.0002$, (Mann–Whitney test). *B–I*, catwalk gait analysis test performed using the CatWalk XT system. Twelve weeks old mice were acclimatized and studied on 3 consecutive days. Each day 3 successful runs were averaged per mouse. *B*, base of support, Front Paws (BOS FP); WT *vs*. dHT, *$P = 0.0126$; dHT *vs*. Ex36, *$P = 0.0148$. *C*, base of support, Hind Paws (BOS HP); WT *vs*. dHT, **$P = 0.0027$; dHT *vs*. Ex36, **$P = 0.0057$. *D*, Stride Length, Right Front Limb (Stride RF), **$P = 0.0089$. *E*, Stride Length, Left Front limb (Stride LF), **$P = 0.0071$. *F*, Stride Length, Right Hind limb (Stride RH), **$P = 0.0048$. *G*, Stride Length, Left Hind limb (Stride LH), *$P = 0.026$. *H*, Duty Cycle, Right Hind (Duty Cycle RH), **$P = 0.0077$. *I*, Duty Cycle, Left Hind (Duty Cycle LH), *$P = 0.039$. Each symbol represents results obtained from a single muscle. WT, $n = 12$ (black symbols); dHT, $n = 7$ (blue symbols) and Ex36 = 6 (red symbols). All statistical analysis were performed using the Mann–Whitney test. Results are normalized to values obtained in WT mice (= 100%).

**Table 4. Analysis of Catwalk parameters in WT, dHT and Ex36 mice. All values are expressed as a percentage of the values obtained in WT mice**

| Parameter | Paw | WT (n = 12) | dHT (n = 7) | Ex36 (n = 6) |
|---|---|---|---|---|
| Stand | RH | 100.0 ± 18.4 | 119.4 ± 26.7 | 134.9 ± 29.3* (P = 0.026) |
| | LH | 100.0 ± 21.8 | 124.1 ± 24.7* (P = 0.026) | 105.8 ± 31.6 |
| Stride Length Front Paws | RF | 100.0 ± 7.6 | 81.2 ± 13.3** (P = 0.0089) | 92.9 ± 13.6 |
| | LF | 100.0 ± 7.7 | 81.1 ± 13.2** (P = 0.0071) | 97.2 ± 13.3 |
| Stride Length Hind Paws | RH | 100.0 ± 11.9 | 79.6 ± 11.5** (P = 0.003) | 87.9 ± 14.6 |
| | LH | 100.0 ± 11.5 | 83.8 ± 15.0* (P = 0.023) | 93.4 ± 15.6 |
| Duty cycle | RH | 100.0 ± 16.6 | 123.2 ± 11.7** (P = 0.0031) | 118.3 ± 22.3 |
| | LH | 100.0 ± 16.5 | 117.6 ± 12.9* (P = 0.039) | 107.9 ± 30.0 |
| BOS | FP | 100.0 ± 12.6 | 81.2 ± 7.2* (P = 0.012) | 102.2 ± 23.3 |
| | HP | 100.0 ± 9.2 | 83.4 ± 6.2* (P = 0.0027) | 107.2 ± 10.7 |
| Minimum intensity | RH | 100.0 ± 14.3 | 95.1 ± 4.8 | 104.7 ± 24.3 |
| | LH | 100.0 ± 12.3 | 88.2 ± 5.7* (P = 0.017) | 102.4 ± 22.7 |
| Standing on three | NA | 100.0 ± 57.1 | 269.2 ± 10.1** (P = 0.0077) | 177.0 ± 111.6 |
| Print area | RH | 100.0 ± 17.1 | 93.5 ± 20.1 | 109.8 ± 21.5 |
| | LH | 100.0 ± 16.6 | 80.1 ± 106.4** (P = 0.0032) | 87.3 ± 35.4 |
| Initial dual stance | RH | 100.0 ± 99.9 | 283.3 ± 128.5* (P = 0.011) | 131.3 ± 104.9 |
| | LH | 100.0 ± 64.0 | 349.1 ± 266.5* (P = 0.010) | 91.3 ± 87.3 |
| Terminal dual stance | RH | 100.0 ± 42.8 | 259.6 ± 200.8* (P = 0.041) | 84.9 ± 64.2 |
| | LH | 100.0 ± 106.2 | 349.7 ± 140.5* (P = 0.010) | 105.22 ± 78.9 |

co-ordination were different in dHT compared to WT littermates (Fig. 6*B–I* and Table 4). For example, within the interlimb co-ordination parameters, dHT mice had a significantly lower Base of Support (BOS, is the average width between footprints during locomotion and is an indication of balance and stability) of front and hind paws (Fig. 6*C*). Specifically, the BOS of the front paws (normalized, mean ± SD) of WT and dHT was 100.0 ± 12.6 *vs.* 81.2 ± 7.2, respectively (P = 0.012). The hind paws BOS (normalized, mean ± SD) of WT and dHT was 100.0 ± 9.2 *vs.* 83.4 ± 6.2, respectively (P = 0.0027). We also analysed the Stride Length, comprising a parameter that detects gait alterations (Kryaiku et al., 2016), by measuring the distance between successive placements of the same paw, measured during

maximal contact. In the stride, both front and hind paws of dHT mice showed a significant decrease compared to those of WT littermates. The stride length, was 20 % lower in the four limbs of dHT mice, Right Front (WT = 100.0 ± 7.6 *vs.* dHT 81.2 ± 13.3; P = 0.0089), Left Front (WT = 100 ± 7.7 *vs.* dHT 81.1 ± 13.2; P = 0.0071), Right Hind (WT = 100.0 ± 11.9 *vs.* dHT 79.6 ± 11.5, P = 0.003) and Left Hind (WT = 100.0 ± 11.5 *vs.* dHT 83.8 ± 15.0; P = 0.023) (Fig. 6*D* and 6*E*). Finally, the Duty cycle defined as percentage of Stand (duration in seconds of contact of a paw with the glass plate) was also significantly higher in dHT mice compared to WT littermates. Specifically, the normalized mean ± SD Duty cycle values of WT *vs.* dHT were: Right Hind (100.0 ± 16.6 *vs.* 123.2 ± 11.7, respectively P = 0.0031) and Left Hind

(100.0 ± 16.5 *vs.* 117.6 ± 12.9, respectively, *P* = 0.039). None of the examined parameters except for the Right Hindlimb stand, which was increased in Ex36 mice (134.9 ± 29.3 *P* = 0.026), differed between WT and Ex36 (Fig. 6*D*). A complete list of all analysed parameters is provided in Table 4. Of importance, the monoallelic expression of RyR1 in Q1970fx12 (Ex36) mice not only failed to result in any gross morphological alterations in spindle morphology or diameters (Fig. 5*B* and *C*), but also did not affect gait and interlimb co-ordination (Fig. 6*B–I*)

In conclusion, the CatWalk analysis results demonstrate that the scores of WT mice are significantly different from those of dHT littermates, supporting our hypothesis that the compound heterozygous RyR1p.Q1970fsX16+p.A4329D mutations not only affect extrafusal muscles, but also may impinge proprioceptive functions controlled by muscle spindles. In the subsequent experiments, we investigated whether the compound heterozygous RyR1p.Q1970fsX16+p.A4329D and *Ryr1* Q1970fx12 (Ex36) mutations affect spine alignment, another indicator of proper proprioceptor function (Blecher et al., 2017).

## dHT mice show skeleton abnormalities

The proprioceptive system is involved in maintaining spinal alignment and indeed, patients with recessive *RYR1 m*yopathies often present several skeleton abnormalities from birth, as well as joint contractures. In addition, skeleton abnormalities occur in murine models lacking Egr3 or RunX3, comprising key proteins that are needed for the proper function of muscle spindle and proprioceptive circuitry, respectively (Blecher et al., 2017). On the basis of these observations, we next analysed by high resolution microtomography whether dHT mice show skeleton abnormalities. Figure 7*A* shows a representative image of the whole-body view of the skeletons of 12-week-old WT, dHT and Ex36 mice. The red line indicating the trajectory of the spinal cord confirm that dHT exhibit scoliosis and the white arrows mark changes of the spinal alignment. To quantify the degree of scoliosis, we measured Cobb angle values, which are the most widely used measurements to quantify the magnitude of scoliosis. Figure 7*B* shows Cobb angle values for the spine of WT, dHT and Ex36 mice and each symbol represents results obtained from a single mouse (*n* = 5, WT; *n* = 5, dHT; *n* = 5, Ex36). The spine of dHT mice have a calculated (mean ± SD) Cobb angle of 19.3 ± 9.4, whereas the spines of WT littermates have mean Cobb angles of 2.4 ± 0.9 (*P* = 0.036). On the other hand, *Ryr1* Q1970fx12 (Ex36) mice showed no significant skeleton alterations; the mean ± SD Cobb angle values for the spine were 2.3 ± 0.9 (*n* = 5 mice) *vs.* 2.6±0.6 (*n* = 5 mice), in

WT and Ex36, respectively (Fig. 7*B*). When examining the skeletons, we also observed marked changes in the spinal alignment leading to kyphosis (Fig. 7*C*) in dHT mice (for the complete dataset of skeleton analysis, see https://zenodo.org/records/12724636).

## Structural disorders of muscle spindle in dHT mice are paralleled by the severe depletion of RyR1 protein and significant alterations of the proteome in intrafusal muscle fibres

Next, we investigated whether the presence of the mutant RyR1s is accompanied by a reduction of RyR1 protein content in the muscle spindles, and whether this is associated with changes of the proteomic signatures. High resolution confocal immunohistochemical analysis on cryosections of EDL and soleus muscles from dHT-Thy-EYFP mice (Fig. 8*A* and *B*) did not reveal any major

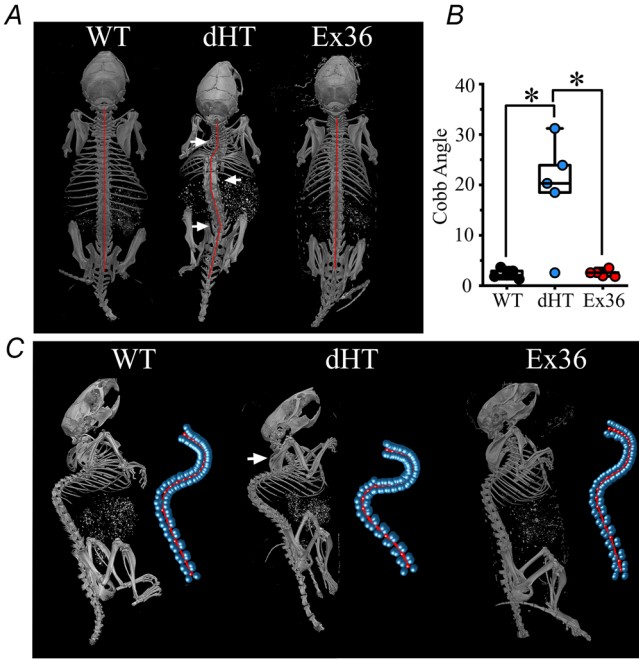

**Figure 7. dHT mice but not Ex36 mice, show skeletal deformities, including scoliosis and kyphosis**
Representative mCT imaging of skeletons from 12-week-old WT, dHT and Ex36 mice. *A*, top view of the whole-body skeleton of WT (left), dHT (central) and Ex36 (right) littermates. The red lines follow the spinal columns and the white arrows mark changes in the spinal alignment. *B*, whisker plots of Cobb angles. Each symbol represents the Cobb value from a single mouse (WT, *n* = 5, black symbols; dHT, *n* = 5, blue symbols; Ex36, *n* = 5, red symbols) WT *vs.* dHT, *\*P* = 0.012; dHT *vs.* Ex36, *\*P* = 0.013, Mann–Whitney test). *C*, lateral views of whole-body skeletons from WT (left), dHT (central) and Ex36 (right) mice. A 3-D reconstruction of the spine is shown on the right of each mouse. There are eight blue spheres for each vertebra, and the red central line follows the path between the vertebras. Kyphosis is very pronounced in dHT mice (white arrow).

changes in the appearance of the nuclear clusters of the equatorial region of intrafusal muscle fibres as indicated by DAPI staining. Similar to that observed in WT-Thy-EYFP mice, the RyR1 Ab preferentially stains the polar regions of the intrafusal fibres. RyR1 immunostaining of the intrafusal fibres from dHT shows the double row pattern, typical of the calcium release units of the junctional sarcoplasmic reticulum.

Because immunohistochemistry is not a quantitative method, we compared RyR1 protein content in muscle spindles from soleus and EDLs isolated from WT, dHT and Ex36 mice by high resolution mass spectrometry. Figure 9*A* shows that the RyR1 protein content in intrafusal muscle fibres of dHT mice is decreased by 54.7% ($P = 0.00216$) compared to that of WT mice. On the other hand, there was no significant difference of the RyR1 protein content between intrafusal fibres from WT and Ex36 mice. As depicted in the volcano plot in Fig. 9*B*, the protein composition of intrafusal fibres from dHT mice differs from that of WT mice with >30 proteins showing significant changes after filtering for those proteins showing ≥2 peptides, exhibiting a fold change ≥0.20 ($-0.321 \leq \log_2 FC \leq 0.263$) and $q \leq 0.05$. In total, 14 proteins were up-regulated and 24 proteins were down-regulated in intrafusal muscles from dHT *vs.* WT. Table 5 shows the complete list of up-regulated and down-regulated proteins. The most affected pathways are those involved in signalling regulation, cellular amide biosynthesis and cellular organization that are significantly increased in dHT *vs.* WT intrafusal fibres (Fig. 9*C*). Pathways involved in cell metabolism, biosynthesis and peptide metabolism are significantly down-regulated in dHT intrafusal fibres (Fig. 9*D*). The proteins showing the greatest enrichment in intrafusal fibres from dHT *vs.* WT mice are: (i) prolyl4-hydroxylase subunit $\alpha$2 a subunit of a key enzyme involved in the proper three-dimensional folding of newly synthesized procollagen chains; (ii) prepl prolyl endopeptidase a ptolyl endopeptidase-like enzyme; and (iii) transferrin receptor (Table 5). Of interest, stromal interaction molecule 1 (or Stim1) a transmembrane protein mediating calcium influx after calcium store depletion, was increased by 1.43-fold ($q = 0.041$) in dHT *vs.* WT intrafusal fibres. On the other hand the proteins showing the largest decrease in dHT intrafusal fibres are $\beta$-IGH3 a TGF-$\beta$ inducible protein implicated in connecting matrix molecules to each other and facilitating cell–collagen interactions, NAD(P)H quinone dehydrogenase 1, a

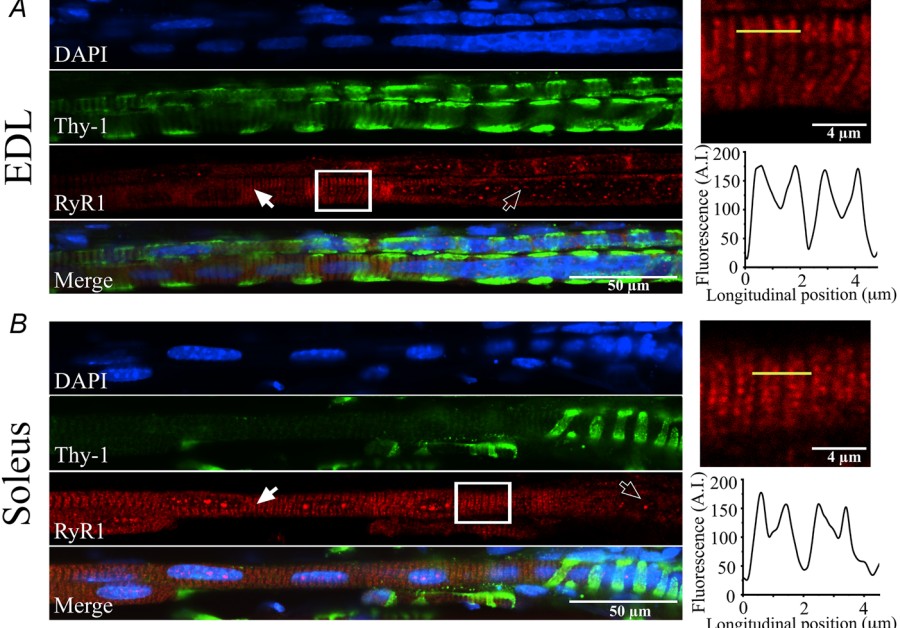

**Figure 8. Immunohistochemical and proteomic analysis of intrafusal muscle fibres in dHT mice**
*A* and *B*, subcellular localization of RyR1 protein in muscle spindle fibres from 12-week-old dHT mice, Confocal images were acquired along the longitudinal axis of muscle spindle fibres from EDL (*A*) and soleus muscles (*B*). Images show the general structure of the muscle spindle. In the polar region (white arrows), the RyR1 positive immunofluorescence is highest. Fibres were fixed and processed as described in the Methods, and stained with DAPI (blue), Thy-1 (green) and RyR1 (red). Images were acquired using a Nikon AxR confocal microscope equipped with a 40× Plan Apo VC Nikon objective (N.A. = 0.95). The two insets show a higher magnification of the boxed region in EDL and soleus RyR1 staining, where the double labelling pattern characteristic of RyR1 distribution is clearly visible. The yellow bar shows the longitudinal position of the fluorescence labelled RyR1. The lower panels of the two insets show the longitudinal spatial fluorescence profiles over the transverse direction within the line.

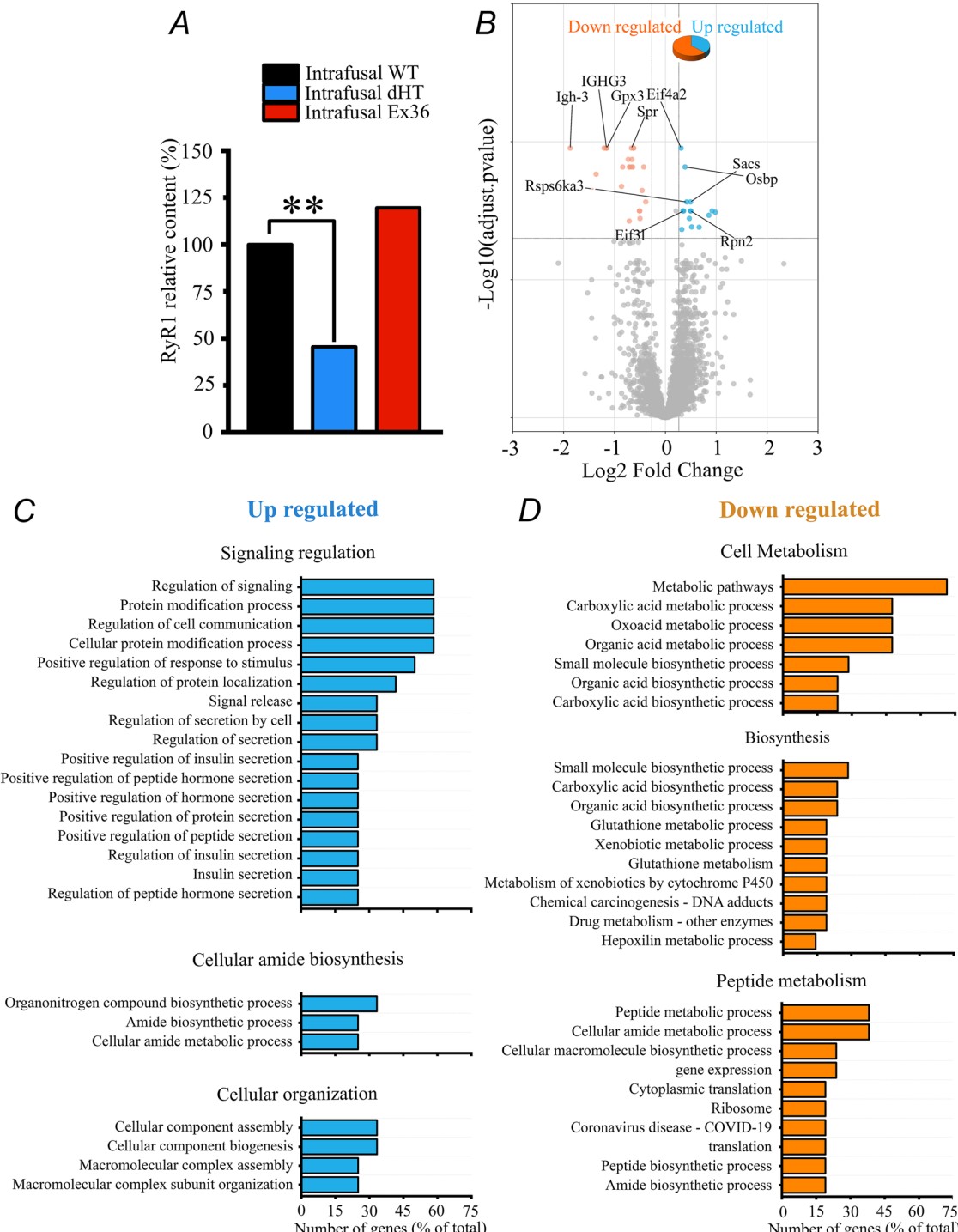

**Figure 9. Proteomic analysis of intrafusal muscle fibers in dHT versus WT mice**
*A*, mass spectrometry quantification shows that the RyR1 protein in intrafusal fibres from dHT mice (isolated from EDL and soleus muscles) is decreased by 54.7% compared to WT littermates. Results are the mean of muscle spindles from WT, *n* = 5; dHT, *n* = 7; Ex36, *n* = 5) WT *vs*. dHT, **P = 0.00216; dHT *vs*. Ex36, ***P = 0.00013. *B*, volcano plot comparison of proteins in muscle spindles from WT *vs*. dHT littermates. After filtering data stringently [keeping only proteins having ≥2 peptides, exhibiting a fold change ≥ 0.20 (–0.321 ≤ log$_2$FC ≤ 0.263) and *q* ≤ 0.05], 24 proteins were down-regulated and 14 up-regulated in spindles from WT *vs*. dHT, respectively. *C*, GO analysis of proteins showing a significant enrichment in dHT *vs*. WT intrafusal muscles. The number of genes annotated to each cluster was calculated as a percentage over the total number of upregulated genes (= 14). *D*, GO analysis of proteins showing a significant depletion in dHT *vs*. WT intrafusal muscles. The number of genes annotated to each cluster was calculated as a percentage over the total number of down-regulated genes.

**Table 5. List of significantly up-regulated and down-regulated proteins in dHT *vs*. WT intrafusal muscles**

| Up-regulated in dHT *vs*. WT | | | Down-regulated in dHT *vs*. WT | | |
| --- | --- | --- | --- | --- | --- |
| Gene name | FC | *q* value | Gene name | FC | *q* value |
| P4ha2 | 1.975703 | 0.032448 | Cbr1 | 0.765177 | 0.027333 |
| Prepl | 1.890943 | 0.031788 | Mdh1 | 0.744469 | 0.015255 |
| Tfrc | 1.808806 | 0.034181 | Nme2 | 0.728192 | 0.022598 |
| Cdk5rap3 | 1.584465 | 0.04171 | Ldhb | 0.70785 | 0.036035 |
| Stim1 | 1.430249 | 0.041519 | Acy1 | 0.705856 | 0.031788 |
| Rpn2 | 1.411756 | 0.031788 | Gatd3 | 0.701152 | 0.031788 |
| Mcu | 1.410518 | 0.031788 | Gstp1 | 0.649432 | 0.011138 |
| Sacs | 1.408841 | 0.027333 | Kyat3 | 0.646493 | 0.011138 |
| Mrpl9 | 1.386239 | 0.036035 | Got1 | 0.640539 | 0.015255 |
| Rps6ka3 | 1.337576 | 0.02735 | Gstm2 | 0.634352 | 0.013451 |
| Osbp | 1.306942 | 0.015255 | Selenbp1 | 0.630464 | 0.015255 |
| Eif3l | 1.282809 | 0.031788 | Spr | 0.630359 | 0.011138 |
| Otud4 | 1.271351 | 0.031795 | Rpl4 | 0.613312 | 0.015255 |
| Iars1 | 1.251002 | 0.043284 | Rpl7 | 0.611864 | 0.015255 |
| | | | Rpl13 | 0.611216 | 0.037633 |
| | | | Gstm1 | 0.602801 | 0.013451 |
| | | | Mb | 0.559002 | 0.015255 |
| | | | Bckdhb | 0.550348 | 0.021111 |
| | | | Gpx3 | 0.449778 | 0.011138 |
| | | | IGHG3 | 0.448368 | 0.011138 |
| | | | Rpl28 | 0.445844 | 0.011138 |
| | | | Nqo1 | 0.434254 | 0.011138 |
| | | | HVM51 | 0.389294 | 0.017207 |
| | | | Igh-3 | 0.273444 | 0.011138 |

cytosolic FAD-binding enzyme involved in the reduction of quinones to hydroquinones and, in the ubiquitin independent p53 degradative pathway, ribosomal protein L28 (Table 5) (for complete mass spectrometry datasets, see https://massive.ucsd.edu/ProteoSAFe/private-dataset.jsp?task = 09ff33af463e430f8e40ce18bf5cf1d2).

### Intrafusal fibres from dHT mice exhibit reduced electrically evoked Ca²⁺ transients

To provide a more direct link between the changes in muscle spindle morphology and altered proprioceptor function observed in the dHT mice caused by altered RyR1-mediated calcium release, we measured electrically evoked $Ca^{2+}$ transients in single isolated intrafusal muscle fibres. Viable contracting intrafusal muscle fibres were dissected from EDL muscles of Thy1-EYFP WT, Thy1-EYFP Ex36 and Thy1-EYFP dHT mice as described in the Methods. In the Supporting information, Video S3 shows the electrically evoked contraction of an isolated intrafusal fibre from a WT mouse. Figure 10 shows a schematic representation of the microscope chamber used to dissect muscle spindles from EDL bundles (Fig. 10*A*) and representative photomicrographs of a dissected intrafusal muscle fibre isolated from a WT mouse at different

low (3× and 6×) magnifications (Fig. 10*B*). Intrafusal fibres are surrounded by an annulospiral Thy1 positive sensory nerve ending, which is positive for YFP-fluorescence (Fig. 10*C*) and contain a centrally located nuclear bag region enclosing many DAPI positive nuclei.

To directly assess whether the function of RyR1 in intrafusal fibres was impacted in dHT mice, we isolated intrafusal fibres from WT, Ex36 and dHT mice as described in the Methods, loaded them with the fluorescent $Ca^{2+}$ indicator Fura-Red (for a representative image, see Fig. 11) and electrically stimulated them by field stimulation with a single pulse of 50 V delivered for 1 ms. Figure 11*B* shows representative traces of electrically evoked $Ca^{2+}$ transients from intrafusal fibres isolated from WT (Fig. 11*B*, left), dHT (Fig. 11*B*, central) and Ex36 (Fig. 11*B*, right) mice and Fig. 11*C* shows the mean ± SD peak changes in fluorescence. The peak $\Delta F$ values observed in intrafusal fibres from WT ($n = 4$ mice) and Ex36 ($n = 4$ mice) were similar (1.31 ± 0.07 and 1.34 ± 0.05, respectively; $P = 0.6076$, ANOVA followed by Tukey's *post hoc* test). On the other hand, the peak $\Delta F$ observed in intrafusal fibres from dHT mice ($n = 4$ mice) was significantly reduced (the mean ± SD $\Delta F$ was 0.99 ± 0.08, $P = 0.00679$, dHT *vs*. WT, and $P = 0.00455$, dHT *vs*. Ex36, ANOVA followed

**Table 6. Analysis of electrically evoked calcium transients in intrafusal muscle fibres from WT, dHT and Exon36 littermates**

| Genotype (number of fibres analysed) | Resting Fura-red fluorescence intensity (a.u.) | $\Delta F/F$ (mean ± SD) | TTP (ms) (mean ± SD) | HTTP (ms) (mean ± SD) | HRT (ms) (mean ± SD) |
|---|---|---|---|---|---|
| WT (n = 4) | 1273 ± 256 | 1.31 ± 0.07 | 11.03 ± 2.28 | 7.57 ± 1.22 | 26.67 ± 4.08 |
| Ex36 (n = 4) | 1299 ± 207 | 1.34 ± 0.05¶¶ | 10.67 ± 1.98 | 7.22 ± 1.74 | 27.89 ± 4.90 |
| dHT (n = 4) | 1308 ± 322 | 0.99 ± 0.08** | 10.57 ± 2.36 | 7.18 ± 1.88 | 28.76 ± 5.67 |

ANOVA Tukey test: **$P < 0.01$ dHT *vs*. WT ($P = 0.00679$); ¶¶ $P < 0.01$, Exon36 *vs*. dHT ($P = 0.00455$)

by Tukey's *post hoc* test) (for the complete dataset, see Table 6).

## Discussion

In the present study, we demonstrate that intrafusal muscles express the RyR1 protein and that *in vivo* parameters associated with proprioceptor functions are affected in dHT mice carrying one hypomorphic allele and one allele containing the p.A4329D missense *Ryr1* mutation. The observed changes in muscle spindle protein composition brought about by the presence of *Ryr1* mutations, causes a massive decrease of RyR1 expression in intrafusal fibres. This, together with reduced electrically evoked Ca$^{2+}$ transients observed in intrafusal fibres from dHT mice, are probably causally linked to

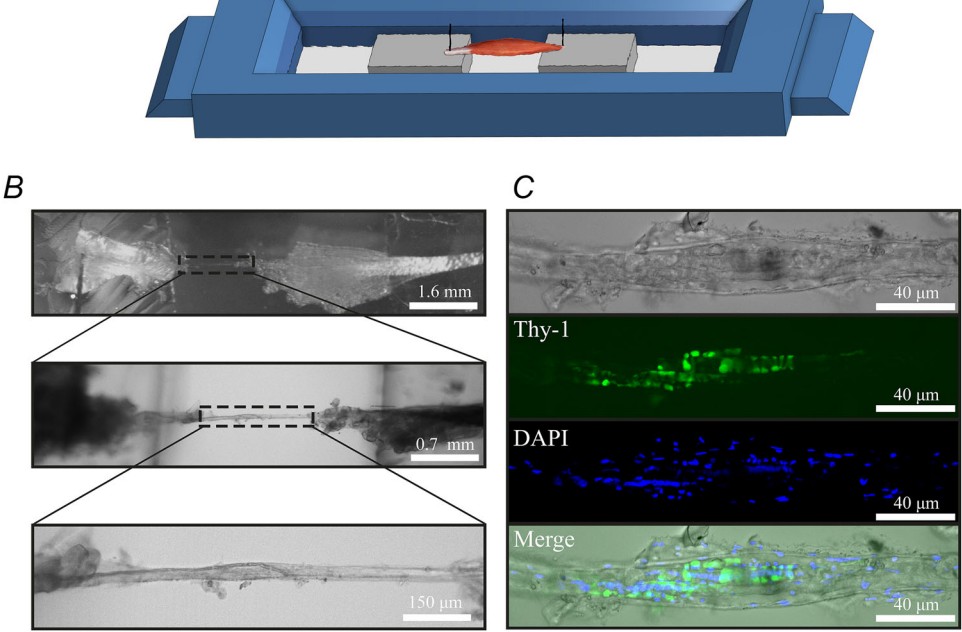

**Figure 10. Isolation of functional muscle spindles**
*A*, schematic representation of the procedure used for muscle spindle isolation from EDL muscles. *B*, low magnification brightfield photomicrographs of an isolated intact muscle spindle (MZ10 F Modular Stereo Microscope; Leica; 3× and 6× magnification). *C*, high magnification confocal images of a muscle spindle. Top image, brightfield. Second image from top, Thy-1-EYFP marking all sensory neuron projections, including group Ia and group II sensory afferent endings in the equatorial region of intrafusal muscle fibres. Third image from top, DAPI fluorescence showing location of nuclear bag region. Bottom image, merged images. Images were obtained with a Nikon A1plus confocal microscope equipped with Nikon Plan Apo, 40× oil objective (N.A. - 1.4) and with Coherent Sapphire (405 nm).

the gait abnormalities observed in dHT mice and to the gross alterations of muscle spindle structure. If these alterations observed in dHT mice also occur in humans, our results could explain at least in part, the skeletal abnormalities, motor delays, joint contractures and gait abnormalities often observed in patients with RYR1 congenital myopathies.

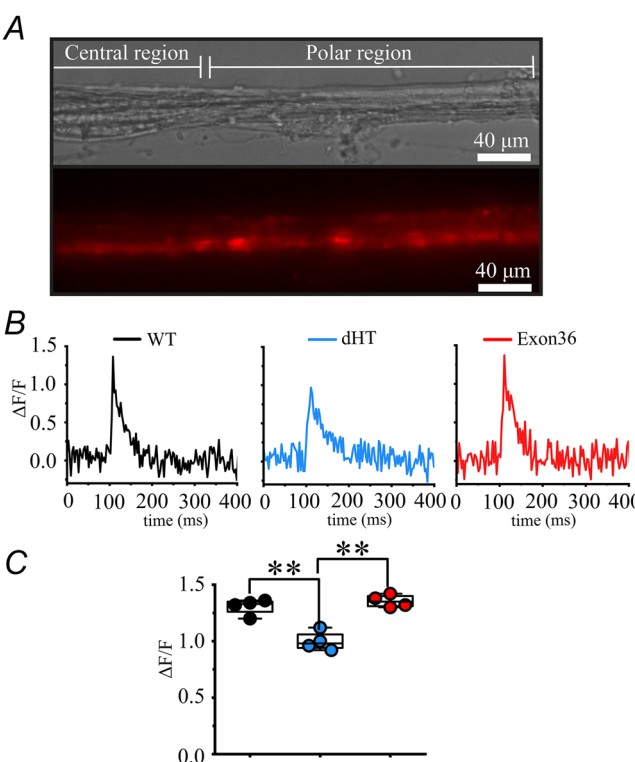

**Figure 11. Intrafusal fibres from dHT mice show reduced electrically evoked calcium transients**

*A*, photomicrograph of a muscle spindle showing the central and polar regions. Top: brightfield. Bottom: the same intrafusal loaded with the fluorescent Ca$^{2+}$ indicator Fura-Red. Images were acquired using a Nikon Eclipse TE2000-E fluorescent microscope equipped with a 20× Plan Apo VC Nikon objective (N.A. = 1.4). Fura-Red was excited at 405 nm (Nikon D-Eclipse C1 laser) and emission was recorded at 655 nm using a Laser Beamsplitter zt 405 RDC filter (AOI 0°, 655/15 BrightLine HC). *B*, representative calcium transients evoked by a single pulse of 50 V (1 ms duration) in Fura-Red loaded intrafusal fibres from WT (black trace), dHT (blue trace) and Ex36 (red trace) mice. *C*, mean ± SD ΔF/F following electrical stimulation of single intrafusal fibres isolated from WT (black symbols), dHT (blue symbols) and Ex36 (red symbols). Fura-Red fluorescence changes were acquired using an electron multiplier CCD camera (C9100-13 Hamamatsu Photonics), using a subarray of 32 × 32 pixels, binning 2 × 2, at a frequency of 300 frames s$^{-1}$ (0.3 kHz). Acquired images were analysed using Metamorph 5.7.4 software; results are reported as $\Delta F/F_0 = (F_{max} - F_{rest})/(F_{rest})$. For experimental details, see Methods. Each dot represents the measurements obtained from a single fibre isolated from the EDL muscle of a single mouse. \*\**P* < 0.01, ANOVA, followed by Tukey's *post hoc* test.

## Muscle spindles show a unique protein composition

We used laser capture microdissection (Espina et al., 2006), to define the proteome of murine intrafusal muscle fibres. Our results demonstrate the validity of our experimental approach because intrafusal muscles are endowed with a unique proteome compared to extrafusal fibres, which are enriched in pathways involved in muscle contracture, sarcomere organization, oxidative phosphorylation and mitochondrial function. In particular, they are enriched in proteins involved in extracellular matrix organization, collagen binding, chromatin assembly and filopodium assembly. Importantly, the present study confirms and extends single nucleus transcriptomics analysis (Kim et al., 2020) and mouse masseter muscle spindle mass spectrometry data (Bornstein et al., 2023). Specifically, we confirm the presence of several proteins identified in murine muscle spindles including Padi2 identified in nuclear Bags (Kim et al., 2020) and within intrafusal fibres (Bornstein et al., 2023; Cicek et al., 2022). Deficiency of Padi2 leads to loss of bone mass by interfering with the transcription factor Runx2 (Cbfa1), comprising an essential master of transcription of skeletogenesis (Blecher et al., 2017; Kim et al., 2023; Komori, 2000).

By mass spectrometry, Bornstein et al. (2023) identified a total of 40 proteins that are uniquely up-regulated in muscle spindles compared to the surrounding extrafusal masseter muscles. Here, we identified more than 1500 up-regulated proteins, some of which are in common with those observed by Bornstein et al. (2023), including Tubb3, Vcan, Eln, Efmp1, Postn, ATP1a3, Hspa12a, Vat1l, Plp1, Myl4; Myl6b and Padi2. We were unable to confirm the enrichment of Slc2a1 encoding the Glut1 transporter, most probably because it is expressed in the outer-most portion of the capsule (Bornstein et al., 2023), an area that was not collected using laser capture microdissection. Of relevance, muscle spindles contain the RyR1 protein, albeit to a much lower extent than that present in extrafusal muscles.

## Alterations of the structure of the spindle in dHT mice

Muscle spindles are composed of intrafusal muscle fibres surrounded by a thin spindle-shaped capsule made up of specialized cells and connective tissue whose role is postulated as being a pressure sensory organ (Bridgeman & Eldred, 1964) and/or a metabolically-active diffusion barrier to the entrance of substances from the external milieu (Ovalle, 1972). The broadness of the gap between muscle spindles and extrafusal fibres is changed in a number of neuromuscular disorders and/or animal models. For example, in muscle spindles of patients with Duchenne muscular dystrophy, the gap between muscle spindles and the extrafusal fibre region becomes

significantly wider (Banks et al., 2005; Karaizou et al., 2007). In the case of neuromuscular disorders linked to *Ryr1* mutations, the molecular basis of gap enlargement appears to be more complex. For example, in a model of central core disease, mice carrying the heterozygous RyR1 p.I4895T mutation exhibited a widening of the gap between the spindles and the myofibres, which became more prominent in aged mice (Zvaritch & MacLennan, 2015). However, our results on Ex36 and HO mouse models suggest that the presence of RyR1 mutations in both intra- and extrafusal fibres is not sufficient to cause changes in the broadness of the gap between muscle spindles. We hypothesize that the increase of the diameter of the gap requires the presence of the *Ryr1* mutation combined with a remarkable decrease of the RyR1 protein content in the intrafusal fibres, as observed in the dHT mice. This is in apparent contrast to the results of Zvaritch & MacLennan (2015); however, they investigated the effect of a dominant loss of function mutation, whereas we investigated the effect of compound heterozygous missense mutations including a hypomorphic allele and a missense mutation in the other allele.

## Alteration of proprioceptive function in dHT mice

We are aware that the Beam walk and CatWalk tests do not directly evaluate only muscle spindle function. Nevertheless, they do appraise proprioceptor parameters including motor co-ordination and balance and have been used for the early detection of motor deficits in mouse models of Huntington's disease, in ALS, Parkinson's disease and Pompe's disease (Brooks & Dunneett, 2009; Vergouts et al., 2015; Wang et al., 2012; Watkins et al., 2023). CatWalk gait analysis revealed a decreased BOS in dHT. The alteration of proprioception can lead to a reduction in the BOS for several reasons related to the body's ability to perceive and respond to its position and movement in space. A reduction of the BOS was also reported in a murine model of Refsum disease (Ferdinandusse et al., 2008), a rare metabolic disease accompanied by ataxia and skeletal abnormalities.

Gait abnormalities might be the result of a variety of factors, including ergogenic deficit of extrafusal fibre, as well as alterations of proprioceptive properties linked to muscle spindle dysfunction. The gait and interlimb co-ordination disorders observed in dHT mice might be the result of several factors, including (i) deficit of muscle strength of extrafusal fibres; (ii) alteration of the contractility of intrafusal fibres upon $\gamma$ motor neuron-dependent stimulation caused by reduced RyR1-mediated calcium release; or (iii) the combination of both. Although we cannot exactly discriminate between these different possibilities, our results suggest that the deficit of muscle strength of extrafusal fibres *per se*

might not be the only factor that should be taken into consideration to explain gait and interlimb co-ordination abnormalities observed in dHT mice. We suggest that the strong decrease of RyR1 content and the decreased efficiency of excitation contraction coupling (caused by reduced RyR1-mediated calcium release) are involved in the interlimb co-ordination abnormalities observed in dHT mice. This idea is consistent with the observation that the Ex36 mouse model displays an average 25% decrease of specific muscle strength, a 40% and 60 % reduction of the RyR1 protein content in extrafusal soleus and EDL muscles, respectively, and yet shows (i) no major changes of gait and interlimb co-ordination properties; (ii) no decrease of RyR1 protein content in intrafusal fibres; and (iii) no changes in electrically evoked $Ca^{2+}$ transients in intrafusal fibres. The latter results are mind boggling and indicate that there might be different factors regulating RyR1 protein content in intrafusal and extrafusal fibres, which may account for these differences. One possibility that should be taken into consideration is the fibre type composition of muscle spindles, which differs from extrafusal muscle fibres (Kucera et al., 1992). In this context, it should be noted that the reduction of RyR1 protein content in extrafusal fibres of slow twitch soleus muscles in Ex36 mice occurs to a much lower extent compared to that in fast twitch EDL muscles. Although the expression pattern of myosin heavy chain isoforms in intrafusal fibres is heterogeneous not only between different intrafusal fibres, but also within different regions of each intrafusal fibres, the polar region of the intrafusal fibres enriched in sarcoplasmic reticulum membrane expresses mostly slow/cardiac twitch myosin heavy chain (Kucera et al., 1992). Thus, the lack and/or smaller decrease of RyR1 content in intrafusal fibres from *Ryr1* Q1970fx12 (Ex36) mice is consistent, at least in a part, with the fibre type composition of intrafusal fibres, which are prevalently fibres expressing slow/cardiac twitch myosin heavy chain, and thus containing lower amounts of RyR1 protein (Eckhardt et al., 2023).

dHT mice exhibit massive reduction of RyR1 protein content in both intrafusal and extrafusal fibres and, in both cases, the decrease of RyR1 protein is associated with a decrease of the amplitude of the calcium transients and muscle strength (Elbaz et al., 2019a; present study). Based on previous (Eckhardt et al., 2023) and present high resolution tandem mass tag mass spectrometry results, we calculated that the RyR1 tetrameric complex content in intrafusal fibres from dHT is 0.071 µmol kg$^{-1}$ wet weight, a value which is 45% of that present in WT intrafusal fibres. The massive reduction of RyR1 protein content in intrafusal fibres from dHT mice is much larger compared to that of Ex36, and we consider that this event would cause a decrease of intrafusal contractility mediated by the smaller calcium transient occurring during $\gamma$ motor neuron stimulation. This in turn could impair the

continuous adjustment of muscle spindle activity during normal locomotor activity. If this is the case, then the deficit of the muscle spindle adjustment occurring in dHT mice is a potential component that should be taken into account when considering the deficit of its proprioceptive system leading to a variety of effects including scoliosis. In this context, it should be noted that patients with some recessive *RYR1* mutations often show scoliosis and joint contractures from birth. Indeed, in a recent retrospective study of 69 patients with *Ryr1* mutations, Sarkozy et al. (2023) showed that 71% patients with AR mutations presented signs at birth, 19% had contractures including contractures of the neck, shoulder, elbow, wrists, thumb, hip, knee and ankles at birth, 23% had scoliosis, and 29% had spinal rigidity (Sarkozy et al., 2023). Similarly, Amburgey et al. (2013) reported that 100% of the patients with recessive *Ryr1* mutations presented symptoms and of these, 50% had scoliosis and 71% contractures that which appeared at birth, before the development major motor milestones such as deambulation.

### The proteomic composition of muscle spindles from dHT mice is altered

Because the expression of RyR1p.Q1970fsX16+p.A4329D mutations causes significant changes in the protein composition of extrafusal muscles (Eckhardt et al., 2023), we were interested in investigating the proteome of intrafusal muscles of dHT mice. The proteome of dHT intrafusal fibres shows that a total of 14 proteins are up-regulated and 24 proteins are down-regulated in dHT compared to WT mice. Interestingly, 21% (5/24) of the down-regulated proteins can either bind to, or are involved in, glutathione metabolism (carbonyl reductase 1, glutathione *S*-transferase pi 1, glutathione *S*-transferase Mu 1 and 2, and glutathione peroxidase 3) and 21% are ribosomal constituents. Down-regulation of proteins involved in glutathione metabolism could lead to protein carbonylation, oxidative stress and mitochondrial dysfunction affecting the function of many proteins, including the RyR1, a major target of oxidative stress (Andersson et al., 2011). Ribosomal proteins are principally involved with the stabilization of the ribosomal subunits, rRNA processing and RNA folding, etc.; however, they also possess extra-ribosomal functions including regulation of cell growth, proliferation, DNA repair and more. Thus, a decrease in large subunit ribosomal proteins content may be linked to altered mRNA processing and/or protein synthesis within the muscle spindles. In the context of RYR1-related congenital myopathies the finding that RPL13 is significantly down-regulated in dHT muscle spindles may be of relevance. Indeed, mutations in RPL13, have been

identified in patients with Spondyloepimetaphyseal dysplasia, a disorder characterized by short stature, scoliosis and kyphosis, hip dislocation, equinovarus foot (club foot) and other orthopaedic joint conditions (Le Caignec et al., 2019), with a large part of these symptoms being shared by patients carrying recessive *RYR1 m*utations.

In the case of up-regulated genes, the protein showing the largest increase in intrafusal of dHT *vs.* WT is collagen prolyl 4 hydroxylase type 2. Collagen prolyl 4 hydroxylase is capable of hydroxylating the majority of proline sites in type 1 collagen, promoting the formation of the triple helix. Thus, the increased content of collagen prolyl 4-hydroxylase may be a downstream effect linked to the abnormal content of collagen in intrafusal muscles fibres from dHT mice.

## Conclusions

In the present study, we provide unambiguous evidence for the expression of RyR1 in intrafusal muscle fibres and demonstrate that the massive RyR1 reduction associated with recessive *Ryr1* mutations, together with the presence of a missense mutation, impacts gait and inter-limb co-ordination and causes significant dysregulation of protein expression. These effects may be involved in the skeleton abnormalities observed in patients affected by recessive *RYR1* mutations. One limitation of the present study is that, although reduced $Ca^{2+}$ transients suggest impaired spindle function, direct measurements of force generation or muscle contraction were not obtained. A second limitation is that we cannot determine whether the altered function of the RyR1 calcium channel is the only cause of the muscle spindle changes or whether these are secondary to changes of protein expression brought about by the presence of the mutations.

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

## Additional information

### Data availability statement

All data, code and materials used in the analysis are available in some form to any researcher for purposes of reproducing or extending the analysis. There are no restrictions on materials, such as materials transfer agreements (i.e. MTAs). All data are available in the main text or the supplementary materials. The Mass spectrometry data has been deposited in the ProteomeXchange Consortium via the ProteoSAFe repository (http://massive.ucsd.edu/ProteoSAFe) with the following access number PXD054222 (https://massive.ucsd.edu/ProteoSAFe/private-dataset.jsp?task = 09ff33af463e430f8e40ce18bf5cf1d2). The complete dataset of skeleton analysis can be found at: https://zenodo.org/records/12724636.

### Competing interests

The authors declare that they have no competing interests.

### Author contributions

F.Z. and S.T. were responsible for conceptualization. A.R., C.H., F.Z., G.S., H.M., K.B., R.P., S.B. and S.T. were responsible for methodology. A.R., F.Z., G.S., H.M., K.B., S.B. and S.T. were responsible for investigations. S.T. and F.Z. were responsible for funding acquisition. S.T. and F.Z. were responsible for project administration. S.T., F.Z. and A.R. were responsible for supervision. A.R., F.Z. and S.T. were responsible for writing the original draft. F.Z., S.T., A.R., S.B., M.F., R.P., L.P., F.Z. and F.P. were responsible for reviewing and editing

## Funding

This project was supported by the following granting agencies: Swiss Muscle Foundation (FSRMM) and Swiss National Science Foundation (SNSF) grant number 310 030_212 192 and NeRAB.

## Acknowledgements

We thank Professor Stephan Kröger for constructive criticism and suggestions.

Open access publishing facilitated by Universitat Basel, as part of the Wiley - Universitat Basel agreement via the Consortium Of Swiss Academic Libraries.

## Keywords

congenital myopathy, intrafusal muscle fibres, muscle spindles, mutations, ryanodine receptor 1

## Supporting information

Additional supporting information can be found online in the Supporting Information section at the end of the HTML view of the article. Supporting information files available:

**Peer Review History**
**Supplementary Material**
**Supplementary Material**
**Supplementary Material**

