## [Peer Review History · The Journal of Physiology]

Massive reduction of RyR1 in muscle spindles of mice carrying recessive *Ryr1* mutations alters proprioception and causes scoliosis.

Alexis Ruiz, Sofia Benucci, Hervé Meier, Georg Schulz, Katarzyna Buczak, Christoph Handschin, Rodrigo C G Pena, Susan Treves, and Francesco Zorzato

DOI: 10.1113/JP287832

Corresponding author(s): Susan Treves (susan.treves@unibas.ch)

Review Timeline:

Submission Date:	10-Oct-2024
Editorial Decision:	13-Nov-2024
Revision Received:	30-Jul-2025
Editorial Decision:	26-Aug-2025
Revision Received:	28-Aug-2025
Accepted:	01-Sep-2025

Senior Editor: Karyn Hamilton

Reviewing Editor: Nikki Jernigan

Transaction Report:

Dear Dr Treves,

Re: JP-RP-2024-287832 "Massive reduction of RyR1 in muscle spindles of mice carrying recessive Ryr1 mutations alters proprioception and causes scoliosis" by Alexis Ruiz, Sofia Benucci, Hervé Meier, Georg Schulz, Katarzyna Buczak, Christoph Handschin, Rodrigo C G Pena, Susan Treves, and Francesco Zorzato

Thank you for submitting your manuscript to The Journal of Physiology. It has been assessed by a Reviewing Editor and by 2 expert referees and we are pleased to tell you that it is potentially acceptable for publication following satisfactory major revision.

LANGUAGE EDITING AND SUPPORT FOR PUBLICATION: If you would like help with English language editing, or other article preparation support, Wiley Editing Services offers expert help, including English Language Editing, as well as translation, manuscript formatting, and figure formatting at www.wileyauthors.com/eoo/preparation. You can also find resources for Preparing Your Article for general guidance about writing and preparing your manuscript at www.wileyauthors.com/eoo/prepresources.

REVISION CHECKLIST:

We look forward to receiving your revised submission.

Yours sincerely,

Karyn Hamilton
Senior Editor
The Journal of Physiology

REQUIRED ITEMS

- Author photo and profile. First or joint first authors are asked to provide a short biography (no more than 100 words for one author or 150 words in total for joint first authors) and a portrait photograph. These should be uploaded and clearly labelled together in a Word document with the revised version of the manuscript. See Information for Authors for further details.

- You must start the Methods section with a paragraph headed Ethical approval (https://jp.msubmit.net/cgi-bin/main.plex?form_type=display_requirements#methods).

Research must comply with The Journal's policies regarding animal experiments (<https://physoc.onlinelibrary.wiley.com/hub/animal-experiments>) and adherence to these policies must be stated in the manuscript.

Authors should confirm in their Methods section that their experiments were carried out according to the guidelines laid down by their institution's animal welfare committee, including an ethics approval reference number. The Methods section must contain a statement about access to food, water and housing, details of the anaesthetic regime: anaesthetic used, dose and route of administration, and method of killing the experimental animals.

- The reference list must be in alphabetical order, rather than numbered, to comply with our Journal format.

- The Journal of Physiology funds authors of provisionally accepted papers to use the premium BioRender site to create high resolution schematic figures. Follow this link and enter your details and the manuscript number to create and download figures. Upload these as the figure files for your revised submission. If you choose not to take up this offer, we require figures to be of similar quality and resolution. If you are opting out of this service to authors, state this in the Comments section on the Detailed Information page of the submission form. The link provided should only be used for the purposes of this submission. Authors will be charged for figures created on this premium BioRender account if they are not related to this manuscript submission.

- Please upload separate high-quality figure files via the submission form.

- Please ensure that the Article File you upload is a Word file.

- Your paper contains Supporting Information of a type that we no longer publish, including supplementary tables and figures. Any information essential to an understanding of the paper must be included as part of the main manuscript and figures. The only Supporting Information that we publish are video and audio, 3D structures, program codes and large data files. Your revised paper will be returned to you if it does not adhere to our Supporting Information Guidelines.

- Papers must comply with the Statistics Policy: https://jp.msubmit.net/cgi-bin/main.plex?form_type=display_requirements#statistics.

In summary:

- If $n \leq 30$, all data points must be plotted in the figure in a way that reveals their range and distribution. A bar graph with data points overlaid, a box and whisker plot or a violin plot (preferably with data points included) are acceptable formats.

- If $n > 30$, then the entire raw dataset must be made available either as supporting information, or hosted on a not-for-profit repository, e.g. FigShare, with access details provided in the manuscript.

- 'n' clearly defined (e.g. x cells from y slices in z animals) in the Methods. Authors should be mindful of pseudoreplication.

- All relevant 'n' values must be clearly stated in the main text, figures and tables.

- The most appropriate summary statistic (e.g. mean or median and standard deviation) must be used. Standard Error of the Mean (SEM) alone is not permitted.

- Exact p values must be stated. Authors must not use 'greater than' or 'less than'. Exact p values must be stated to three significant figures even when 'no statistical significance' is claimed.

- Please include an Abstract Figure file, as well as the Figure Legend text within the main article file. The Abstract Figure is a piece of artwork designed to give readers an immediate understanding of the research and should summarise the main conclusions. If possible, the image should be easily 'readable' from left to right or top to bottom. It should show the physiological relevance of the manuscript so readers can assess the importance and content of its findings. Abstract Figures should not merely recapitulate other figures in the manuscript. Please try to keep the diagram as simple as possible and without superfluous information that may distract from the main conclusion(s). Abstract Figures must be provided by authors no later than the revised manuscript stage and should be uploaded as a separate file during online submission labelled as File Type 'Abstract Figure'. Please also ensure that you include the figure legend in the main article file. All Abstract Figures should be created using BioRender. Authors should use The Journal's premium BioRender account to export high-resolution images. Details on how to use and access the premium account are included as part of this email.

Reviewing Editor:

Comments to ensure the paper complies with the Statistics Policy:

Authors need to provide precise p-values

Comments to the authors:

The paper examines muscle spindle function in the context of recessive RyR1 mutations and presents an intriguing and potentially impactful contribution to the field. However, the primary concern is the lack of direct evidence demonstrating functional impairment of the muscle spindle. Without such data, the authors' conclusion that deficient spindle function contributes to RyR1-related locomotor, gait, and skeletal defects remains somewhat speculative. Establishing a clear connection between altered receptor function and physiological dysfunction is critical for validating the proposed mechanistic insights.

Senior Editor:

Comments to ensure the paper complies with the Statistics Policy:

Please revisit The Journal's statistics policy to ensure compliance. Your Statistical Analyses section in the Methods is quite brief. If you could make sure it is complete and includes indication that you represent your data with means \pm SD in this section, then you can streamline the reporting of your results by not having to repeat this each time. I see that you frequently provide precise p-values, in compliance with The Journal's policy (thank you). We would appreciate it if you can do this in the figures as well. Thank you!

Comments to the authors:

Thank you for submitting your manuscript for consideration by The Journal of Physiology. As part of the peer review process, we recruited two Referees with expertise in this field of study. Both were quite enthusiastic about the work, indicating that they believe that, with revision, this manuscript has the potential to be quite influential. If you believe that you can address the major concerns raised by the referees, we would like to invite you to respond point-by-point to each of the critiques, making the major manuscript revisions required to address them. I've made some additional comments about adherence to The Journal's statistics policy and The Journal Staff may be providing guidance for reorganizing your sections into the more traditional order, with Introduction followed by Methods, Results, and Discussion. Please resubmit the revised manuscript for our continued consideration. Thank you for your interest in The Journal of Physiology!

Referee #1:

Authors investigated how a recessive RyR1 (Ryanodine receptor type 1) mutation affects the function of the muscle spindle. To this end, they first confirmed the presence of this SR (sarcoplasmic reticulum) calcium release channel in intrafusal fibres. They then described a decreased expression of this protein in the dHT mouse strain together with altered gait properties and spinal deformities (scoliosis and kyphosis). They conclude that the often observed skeletal deformities and altered muscle function in patients with recessive mutations in the Ryr1 gene could - at least partially - due to the modified function of the muscle spindle and associated proprioception.

Authors used laser-assisted microdissection to assess the protein expression profile of intrafusal fibres, confocal microscopy to show the presence and localization of RyR1 in these cells, and the Catwalk system to analyse the gait properties of the dHT mice.

Questions and comments:

General:

Authors give a detailed description of the altered protein expression (Fig.6D) in the intrafusal fibres of dHT mice. However, in my opinion, they do not explain how these changes would result in an altered function of the muscle spindle as a sensory organ. They should be more specific in how any of these changes (except the reduction in RyR1 content) presented in Fig.6 could result in an altered contractile response of the intrafusal fibres and a consequently modified action potential profile of the sensory nerve axon. Some direct tests on the functioning of the muscle spindle in dHT mice under anesthetised conditions would be appropriate.

Although discussed to some extent, it is not clear why should the altered RyR1 content and the assumed - but not proven - modification in receptor function be the main underlying reason for the observed changes in the Catwalk parameters. Wouldn't the reduced force (and the smaller calcium transient in the extrafusal fibres; as given in the Suppl. material) be "enough" to explain all the observed changes?

Regarding the Methods, Authors should give an estimate of the extent of nerve contamination (afferent and efferent nerve endings) in their muscle spindle preparation. This would help the Readers interpret the proteomic data.

Other:

page 16:

- It is unclear to the reader why Catwalk parameters should be normalized to body weight. Wouldn't body size be more appropriate?

- Data presented in the text are simple repetitions (except for the actual values of "p" and the averages) shown in Fig. 4. Furthermore, it is not necessary to mention the statistical test used for every comparison as the Authors used the same - Mann Whitney - test in all cases. The text should thus be shortened.

- BOS is not described correctly. Please rephrase. (Maybe "average distance of paws".)

page 30:

- The inserted mathematical symbols are misplaced rendering the text extremely hard to read. Please correct. e.g. "where t is a transversal parameter in the $[0,1]$ interval..."

- I suggest using "first" and "second derivatives" of $\gamma(t)$ instead of "velocity" and "acceleration", as this is not a distance vs. time ($s(t)$) curve where velocity and acceleration have a physical meaning.

Suppl. Fig. 4:

- Although one can deduce it from the Fig. legend, blue circles are not explained in the Figure itself.

Minor

Some typos should be corrected:

- page 6, line 7 should read "Stim1, Orai1, and calsequestrin 1"
 - page 11, 3rd line from bottom; a parenthesis is missing from the end of the sentence
 - page 15, line 6 should read "not in those carrying"
-

Referee #2:

In this work, authors hypothesize that disease mutations in the type 1 ryanodine receptor (RyR1) not only impair the function of extrafusal muscle fibres but also of intrafusal fibres in the muscle spindles, resulting in proprioception deficits, altered gait and skeletal deformities in patients. Authors use a double knock-in mouse model of human recessive RYR1 compound mutations, the dHT mouse and two other RyR1 mouse disease models of milder phenotype. Extrafusal and intrafusal muscle fibres are found to present a differing protein composition in WT mice. RyR1 is present in the intrafusal fibres with a spatial organization similar to that in extrafusal fibres. Muscle spindle morphology differs between WT and dHT mice. dHT mice present skeletal deformities and altered proprioceptive function in behavioural tests. RyR1 expression is reduced in muscle spindles from dHT mice and the level of expression of numerous other proteins is also altered. Authors suggest that altered contractility of intrafusal fibres, due to reduced RyR1 protein level, may contribute to the skeletal and proprioception deficits.

This is a carefully conducted study, making use of three disease mouse models, combining proteomic analysis, histology and immunofluorescence, behavioural tests and characterization of skeletal defects. The tested hypothesis is sound. Results are consistent with the possibility that reduced RyR1 content in the intrafusal fibres compromises the spindle function and contributes to skeletal defects and altered gait. My main concern is that the function of the spindle was not assessed. Only, a 15% increase in spindle diameter is reported in the dHT mouse model, but there is no evidence that physiological function is impaired. Thus, the conclusion remains, to some extent, speculative.

Major

Discriminating the role of reduced RyR1 content in extrafusal vs intrafusal fibres on skeletal deformities and deficient proprioception will be challenging because both types may be involved, as acknowledged by the authors. And for instance, the fact that foetal muscular movements are important for skeletal morphogenesis favours a role for extrafusal fibres as well.

Electrophysiological analysis of spindle function could make the point raised by the authors much stronger. Acknowledgements mention the help of Stephan Kröger who is the world expert in quantitative assessment of spindle function. Wouldn't it be possible to have spindle function tested in the dHT model?

Authors mention in the abstract that the link between RYR1 mutations, altered muscle spindles and skeletal defects has not been investigated. This is not completely true as there is a report by Zvaritch and Mac Lennan in a mouse model of central core disease, where degeneration of the intrafusal fibres is described and possible role in deficient balance, locomotion and coordination is raised

(Biochem Biophys Res Commun. 2015). It is hard to understand why this paper was not quoted and discussed.

The Ex36 mouse model is described as presenting a milder phenotype and smaller reduction in RyR1 than the dHT mouse. However, in EDL muscle the reduction in RyR1 content in extrafusal fibres is very similar in the two models. Supplementary Fig. 3B shows results for spindle morphology only in the Ex36 soleus muscle, for which the decrease in RyR1 is less than in

the EDL. It would be interesting to know about spindle morphology in the Ex36 EDL muscles and or about the extent of RyR1 reduction in Ex36 intrafusal fibres. This could help support the correlation between strong decrease in RyR1 content and changed spindle morphology.

Proteomic analysis generates lots of data which, for a large part, remains descriptive with no or limited explainable relationship to the studied process (for instance, Figure 2 and text pages 9-10). This would fit better as supplementary data with description focussed only on the most relevant pathways in the disease context.

Other

Looking back into the literature starting from ref 38, I could not find specific clinical information regarding the one severely affected patient from which the dHT mouse model was created. Was the patient suffering from skeletal alterations and gait deficit? Is the information readily available or did I miss it?

In the legend of Fig. 1B and Fig. 6C, it should be indicated which muscles were used.

Page12, description of the Ex36 mouse mentions: These carry a frameshift mutation leading to a downstream premature stop codon causing nonsense mediated RNA decay of the mutant allele

and resulting in $\approx 40\%$ decrease of RyR1 content.

It should be mentioned that this is in soleus muscle.

Page 21: referring to the Ex36 mouse, it is said: yet it shows an approx. 20 % decrease of muscle strength and reduction of RyR1 protein content in extrafusal fibres. This makes it sound as if the RyR1 content is reduced by 20%.

Page 8: intrafusal fibers express an adequate amount of RyR1. A term other than adequate may be more appropriate.

typos:

page 4: Dysregulation of calcium homeostasis in skeletal muscles plays

page 11: leading us to next investigated whether

page 12, legend: from WT and dHT mice in.

page 12: which were used to generate the dHT mice. Mann Whitney), respectively.

page 22: is that we cannot determined whether

END OF COMMENTS

Dear Dr. Karyn Hamilton,
thank you for your helpful comments and positive feedback. We were pleased to hear that our paper entitled “Massive reduction of RyR1 in muscle spindles of mice carrying recessive Ryr1 mutations alters proprioception and causes scoliosis“ by Ruiz et al. is potentially acceptable for publication in *Journal Physiology*. We have made major revisions to our manuscript following the Referees comments and below you will find our point by point replies to the Editors as well as Reviewers.

REQUIRED ITEMS:

- The revised Methods section now starts with a paragraph headed Ethical approval where it is stated that we adhere to *J. Physiol.* policies regarding animal experiments.
- The revised Methods section includes the permit numbers, anesthetics used, and housing details.
- The reference list is in alphabetical order.
- High quality and high resolution image files have been uploaded separately.
- The supporting figures and tables have been removed and are mostly incorporated in the new revised figures. The Supplementary information files contain 3 videos.
- All figures include single data points (one symbol = one mouse) as well as bar graphs/whisker plots showing standard deviations.
- The “n” are defined in the Methods section, in the Results section as well as in the Figure legends and in the Tables.
- Exact p values are given even when no statistical significance was found.
- An Abstract Figure file and its legend have been added.

REVIEWING EDITOR:

- Precise p values are given throughout the manuscript.
- We are aware that our manuscript does not provide direct evidence demonstrating that the functional impairment of the muscle spindles causes defects in gait, skeleton abnormalities and gait. In order to provide a more direct link we have measured calcium changes in the intrafusal fibers of WT, Ex36 and dHT mice. The new results (see new Figures 10 and 11, new version of the manuscript) confirm that electrically evoked calcium transients are significantly reduced in intrafusal fibers from dHT mice versus WT and Ex36 mice thereby providing direct evidence of functional impairment in intrafusal fibers from dHT mice

SENIOR EDITOR:

- The statistical analysis section has been re-written and now includes a sentence stating that all values are the mean \pm SD and that the statistical analysis was performed using the Mann-Whitney test, unless otherwise specified.
- Exact p values have also been included in the figure legends.

We would like to thank the Reviewers for their constructive criticisms and hope that in its present version our paper is now acceptable for publication in the *J. Physiol.*

Best regards,

Prof. Francesco Zorzato, M.D., PhD and Prof. Susan Treves, PhD

Reply to the Reviewer's comments

REFEREE 1

1) Authors give a detailed description of the altered protein expression (Fig.6D) in the intrafusal fibres of dHT mice. However, in my opinion, they do not explain how these changes would result in an altered function of the muscle spindle as a sensory organ. They should be more specific in how any of these changes (except the reduction in RyR1 content) presented in Fig.6 could result in an altered contractile response of the intrafusal fibres and a consequently modified action potential profile of the sensory nerve axon.

We have gone over the Mass Spectrometry data to verify if other proteins involved in muscle contraction are significantly changed in intrafusal muscle fibers from dHT vs WT mice. Table 5 shows that two proteins directly involved in calcium regulation are up-regulated in dHT mice, these being Stim 1 and Mcu. However, their relationship if any, to changes in muscle contractility is difficult to envisage and remains to be proven. We speculate that an increase of Stim1 might cause an increase of calcium entry during EC coupling (phasic SOCE) to compensate for lower calcium release via RyR1. However, cytosolic calcium overload by enhanced calcium entry may lead to mitochondrial stress and activation of calpains within intrafusal fibers, which might indirectly affect fiber contractility. Nevertheless, in our opinion, it is very difficult to establish the exact impact of these changes on intrafusal muscle contraction.

When analyzing the changes in content of selected proteins, the only protein directly involved in muscle contraction showing a significant p value ($p=0.006$) was skeletal muscle actin (*Acta1*), whose content was found to be 1.5-fold higher ($\log_2 FC=0.61$) in dHT vs WT and thus unlikely to cause an altered contractile response. For these reasons, we believe that the strong reduction of RyR1 protein together with the decreased RyR1-mediated calcium release in the intrafusal fibers of dHT mice (see Fig.11, revised manuscript) are a plausible cause of the altered contractile response.

2) Some direct tests on the functioning of the muscle spindle in dHT mice under anesthetised conditions would be appropriate.

We think that the activation of the EC coupling of intrafusal fibre via gamma motor neuron activation would be a direct test of spindle function. However, at this point in time it is very difficult, if not impossible, to perform such experiments to directly assess whether changes of EC coupling of intrafusal fibres cause a modification of the action potential in afferent sensory axon neurons. However, we have directly assessed RyR1 function by measuring electrically evoked calcium release, which is the downstream function of the EC coupling mechanism (see Fig.11, revised manuscript).

3) Why should the altered RyR1 content and the assumed - but not proven - modification in receptor function be the main underlying reason for the observed changes in the Catwalk parameters. Wouldn't the reduced force (and the smaller calcium transient in the extrafusal fibres; as given in the Suppl. material) be "enough" to explain all the observed changes?

A great deal of data has shown that reduction of RyR1 content in muscle fibers is associated with a decrease of calcium transients in a variety of experimental models, including our RyR1 knock in mice. Given the ultrastructural similarity of junctional SR between extrafusal and intrafusal fibers, it is difficult to envisage that the reduction of RyR1 content in intrafusal fibers has zero functional effect of calcium release from SR and indeed we have now directly measured electrically evoked calcium transients in intrafusal fibers from WT, Ex36 and dHT mice (Fig.11, revised manuscript).

Furthermore, this suggestion is consistent with the results we obtained with the Ex36 mouse model (Elbaz et al, 2019. Quantitative reduction of RyR1 protein caused by a single-allele frameshift mutation in RYR1 ex36 impairs the strength of adult skeletal muscle fibers. *Hum Mol Genet* **28**, 1872-1884) which displays an average 20 % decrease of specific extrafusal muscle strength, reduction of the RyR1 protein content in extrafusal fibres but yet shows no changes of gait and interlimb coordination properties. Analysis of the Mass spectrometry data presented in the revised manuscript shows that the lack of changes of gait and interlimb coordination is paralleled by an absence of reduction of RyR1 content in the intrafusal fibres of Ex36 mice (Fig. 9, revised manuscript). Additionally, we measured calcium transients in intrafusal fibres from WT, Ex36 and dHT mice and found that the action-potential induced calcium release in intrafusal fibres from dHT mice is lower than that observed in WT and Ex36 mice, supporting the hypothesis that the altered contractile activity of intrafusal fibres might affect the adjustment of muscle spindle afferent firing during locomotor activity.

Collectively, these data are consistent with the idea that the reduction of RyR1 signalling in intrafusal fibres of dHT mice may contribute to the impairment of the continuous adjustment of muscle spindle activity during normal locomotor activity.

4) Authors should give an estimate of the extent of nerve contamination (afferent and efferent nerve endings) in their muscle spindle preparation. This would help the Readers interpret the proteomic data

We found that the following proteins namely: $\alpha 3$ subunit Na⁺/K⁺ ATPase (*Atp1a3*; is present in both afferent and efferent neurons and in the membranes of spiral ganglion somata), dihydropyrimidinase Like 3 (*Dpysl3*; is present in both afferent and efferent nerves, especially in motor neurons, heat shock protein 70 kDa protein 12A (*Hspa12a*; is present in both afferent and efferent neurons as well as other cellular compartments), synaptic vesicle amine transport 1 (*Vat1*; is primarily located in efferent nerves, specifically in the nerve terminals of cholinergic neurons involved in the release of acetylcholine. It's an integral membrane protein of cholinergic synaptic vesicle), proteolipid protein 1 (*Plp1*; is the major myelin protein of the CNS and therefore is present in both afferent and efferent neurons) and Class 3 β -tubulin (*Tubb3*; neither afferent nor efferent but rather it plays a structural and functional role in nerve cells) are enriched in muscle spindles compared to extrafusal fibers (Table 3). Our data are in agreement with those of Bornstein et al. 2023, and suggest that the samples we obtained by laser capture microdissection included proteins associated with nerve endings. We included a statement in this regard, in the revised manuscript (page 16, lines 1-4).

5) Why are Catwalk parameters normalized to body weight. Wouldn't body size be more appropriate?

X-ray microtomography showed that the dHT mouse model shows significant postural changes (hunched back), which would interfere with body size measurements. In addition, Machado et al. (eLife 2015;4:e07892) showed that the stride length parameter is affected by body weight. On the basis of these observations, we decided to normalize the Catwalk parameters to body weight.

6) Data presented in the text are simple repetitions (except for the actual values of "p" and the averages) shown in Fig. 4. Furthermore, it is not necessary to mention the statistical test used for every comparison as the Authors used the same - Mann Whitney - test in all cases. The text should thus be shortened.

We amended the text as suggested by the reviewer throughout the manuscript.

BOS is not described correctly. Please rephrase. (Maybe "average distance of paws".)

In the revised manuscript the meaning of BOS has been clarified and it is now stated that BOS or base of support, refers to the width between the footprints during locomotion, an indication of balance and stability (page 19, lines 17-18, revised manuscript).

7) Page 30:

- The inserted mathematical symbols are misplaced rendering the text extremely hard to read. Please correct. e.g. "where t is a transversal parameter in the $[0,1]$ interval...

- I suggest using "first" and "second derivatives" of $\gamma(t)$ instead of "velocity" and "acceleration", as this is not a distance vs. time ($s(t)$) curve where velocity and acceleration have a physical meaning.

We amended the text as suggested by the reviewer: page 12 top paragraph, the revised manuscript

For Kyphosis, a Permutation statistical test, cubic splines were fit to the 3D point clouds of each mouse in the dataset (<https://zenodo.org/doi/10.5281/zenodo.12721977>). These smooth curves were constrained to be flat (zero curvature) at their extremes, corresponding to the points that would lead to the mouse tail and head connections. That was made to focus on the inner part of the annotated spine columns and avoid spurious curvature at the extremes.

The cubic spline fit leads to a parametrization of the type $\gamma = \gamma(t) = (\gamma_x(t), \gamma_y(t), \gamma_z(t))$ for each spine column, where t is a traversal parameter in the interval $[0,1]$ such that $\gamma(0)$ and $\gamma(1)$ are the

points at the beginning and end of the parametrized smooth curve, respectively. The curvature of a parametrized curve was $k(t)$, computed at each $\gamma(t)$ value of t using the standard formula

$$k(t) = \frac{\|\gamma'(t) \times \gamma''(t)\|}{\|\gamma'(t)\|^3}$$

Where γ' and γ'' are the first and second derivatives γ with respect to t . And $\|\cdot\|$ is the usual Euclidean norm.

The radius of curvature of $\gamma(t)$ at each $t \in [0,1]$ is given by the reciprocal of the curvature:

$$r(t) = \frac{1}{k(t)}$$

We compare populations of mice via summaries of the curvatures of the smooth curves fitted to their spine column point clouds. The code used for the analysis is available at <https://gitlab.com/ceda-unibas/spine-curvatures>

8) Suppl. Fig. 4:

- Although one can deduce it from the Fig. legend, blue circles are not explained in the Figure itself.

Supplementary Figure 4 has now been incorporated into the new Figure 7 of the revised version. The information regarding black and blue symbols has been added to the revised legend to figure 7.

9) Minor: typos have been corrected

- page 15, line 6 should read "not in those carrying"

We amended the text as suggested by the reviewer: **page 18, Line 19** of the revised manuscript.

Referee 2

1) Discriminating the role of reduced RyR1 content in extrafusal vs intrafusal fibres on skeletal deformities and deficient proprioception will be challenging because both types may be involved, as acknowledged by the authors. And for instance, the fact that foetal muscular movements are important for skeletal morphogenesis favours a role for extrafusal fibres as well.

We agree that it is very difficult to discriminate the exact effect of extrafusal and intrafusal fibers on the observed skeletal deformities. However, in the present

investigation by comparing the different mouse models, namely comparing the results obtained on the WT, Ex36 and dHT mouse models, suggest that the deficit of RyR1 in intrafusal fibers is an important component. The reduction of RyR1 protein content leads to a decrease of electrically evoked calcium release and this could interfere/impair with the continuous adjustment of muscle spindle activity during normal locomotor activity. We believe that the structural and functional defects of RyR1 is an important factor of abnormalities of proprioceptive functions leading to the skeletal deformities and gait abnormalities (see page 25, lines 11-34, page 26, lines 1-4 of the revised manuscript) observed in dHT mice.

2) Electrophysiological analysis of spindle function could make the point raised by the authors much stronger. Acknowledgements mention the help of Stephan Kröger who is the world expert in quantitative assessment of spindle function. Wouldn't it be possible to have spindle function tested in the dHT model?

We agree with the Reviewer's comment but unfortunately it was not possible to have the function of the muscle spindles from our mouse model(s) tested by Stephan Kröger due to his lack of available coworkers and because of time constraints. Furthermore, Stephan Kröger's lab had no possibility to measure the response of muscle spindles upon selective gamma motor neuron activation.

3) Authors mention in the abstract that the link between RYR1 mutations, altered muscle spindles and skeletal defects has not been investigated. This is not completely true as there is a report by Zvaritch and Mac Lennan in a mouse model of central core disease, where degeneration of the intrafusal fibres is described and possible role in deficient balance, locomotion and coordination is raised (Biochem Biophys Res Commun. 2015). It is hard to understand why this paper was not quoted and discussed.

We have now added the reference of Zvaritch and MacLennan to the revised version of the manuscript (page 24, line 21 and 26).

4) The Ex36 mouse model is described as presenting a milder phenotype and smaller reduction in RyR1 than the dHT mouse. However, in EDL muscle the reduction in RyR1 content in extrafusal fibres is very similar in the two models. Supplementary Fig. 3B shows results for spindle morphology only in the Ex36 soleus muscle, for which the decrease in RyR1 is less than in the EDL. It would be interesting to know about spindle morphology in the Ex36 EDL muscles and or about the extent of RyR1 reduction in Ex36 intrafusal fibres. This could help support the correlation between strong decrease in RyR1 content and changed spindle morphology.

We have performed H&E staining of muscle spindles from WT, dHT and Ex36 EDL muscles. As can be seen from the included figure, the appearance of the muscle spindles in EDL muscles is similar to that of soleus muscles shown in Figure 5 revised version of the manuscript. That is there are no gross changes between WT and Ex36, whereas in the dHT mice, the spindles are surrounded by a wider gap between the muscle spindle and the extrafusal fibers compared to those of WT. Since the results are similar in both EDL and soleus muscles, we do not think it is necessary to add the information to the revised manuscript.

Figure legend: H&E staining of muscle spindles from WT, dHT and Ex36 EDL muscles. As shown, the appearance of the muscle spindles in EDL muscles is similar to that of soleus muscles shown in Figure 5 revised version of the manuscript.

As to the change in RyR1 content, in intrafusal muscles from Ex36 mice, RyR1 content was similar to that of WT muscle spindles as determined by Mass spectrometry. While RyR1 content in muscle spindles from dHT mice was reduced by more than 50%. These results are now shown in the new Fig.9 panel A.

5) Proteomic analysis generates lots of data which, for a large part, remains descriptive with no or limited explainable relationship to the studied process (for instance, Figure 2 and text pages 9-10). This would fit better as supplementary data with description focused only on the most relevant pathways in the disease context.

According to the policy of *J. Physiol.* all data needs to be incorporated into the text as they no longer accept supplementary material, aside the supporting videos. See reply to point 5 to the handling Editor's queries.

6) Looking back into the literature starting from ref 38, I could not find specific clinical information regarding the one severely affected patient from which the dHT mouse model was created. Was the patient suffering from skeletal alterations and gait deficit? Is the information readily available or did I miss it?

The patient from whom the mouse model was created was described by Klein et al. 2012 and corresponds to patient 35+ in the supplementary information of their paper. He was first described by Zhou et al. 2007(Brain 130:2024) but he unfortunately died at a very young age so no additional information is available.

7) In the legend of Fig. 1B and Fig. 6C, it should be indicated which muscles were used.

In the new legend to Fig.3B and Fig. 9C we now state that muscle spindles were isolated by laser capture from EDL and soleus muscles.

8) Page 12, description of the Ex36 mouse mentions: These carry a frameshift mutation leading to a downstream premature stop codon causing nonsense mediated RNA decay of the mutant allele and resulting in \approx 40% decrease of RyR1 content. It should be mentioned that this is in soleus muscle.

The sentence has been corrected in the new version of the manuscript (page 17, lines 8-9, revised manuscript).

9) Page 21: referring to the Ex36 mouse, it is said: yet it shows an approx. 20 % decrease of muscle strength and reduction of RyR1 protein content in extrafusal fibres. This makes it sound as if the RyR1 content is reduced by 20%.

The sentence has been corrected on page 17, lines 9-11, of the new version of the manuscript.

10) Page 8: intrafusal fibers express an adequate amount of RyR1. A term other than adequate may be more appropriate.

We have replaced the word “adequate” with “assessable” on page 14, line 21 of the new version of the manuscript.

We would like to thank the Reviewers for their constructive criticisms and hope that in its present version our paper is now acceptable for publication in the J. Physiol.

Best regards,

Prof. Francesco Zorzato, M.D., PhD

Prof. Susan Treves, PhD

Dear Dr Treves,

Re: JP-RP-2025-287832R1 **"Massive reduction of RyR1 in muscle spindles of mice carrying recessiveRyr1 mutations alters proprioception and causes scoliosis."** by Alexis Ruiz, Sofia Benucci, Hervé Meier, Georg Schulz, Katarzyna Buczak, Christoph Handschin, Rodrigo C G Pena, Susan Treves, and Francesco Zorzato

Thank you for submitting your manuscript to The Journal of Physiology. It has been assessed by a Reviewing Editor and by 2 expert referees and we are pleased to tell you that it is acceptable for publication following satisfactory revision.

REVISION CHECKLIST:

Please upload two versions of your manuscript text: one with all relevant changes highlighted and one clean version with no changes tracked. The manuscript file should include all tables and figure legends, but each figure/graph should be uploaded as separate, high-resolution files. The journal is now integrated with Wiley's Image Checking service. For further details, see: <https://www.wiley.com/en-us/network/publishing/research-publishing/trending-stories/upholding-image-integrity-wileys->

image-screening-service

We look forward to receiving your revised submission.

Yours sincerely,

Karyn Hamilton
Senior Editor
The Journal of Physiology

EDITOR COMMENTS

Reviewing Editor:

This is a well-written manuscript investigating the role of RyR1 in muscle spindle function and pathology, which will have a significant impact in the field.

The reviewers have provided a few minor comments that should be addressed to strengthen the manuscript.

With regard to muscle spindle function, the inclusion of electrically evoked Ca^{2+} transients sufficiently supports the conclusion that spindle function is impaired. However, please add a brief statement acknowledging the limitation that, while reduced Ca^{2+} transients suggest impaired spindle function, direct measurements of force generation or muscle contraction were not obtained.

Senior Editor:

Thank you for submitting your revised manuscript for continued consideration by The Journal of Physiology. The Referees are pleased with the careful modifications you made to the paper and only have a few remaining minor points to address. In addition to Referee comments, please clarify the light/dark cycle for housing the animals. At this point, we would like to Provisionally Accept your work, pending satisfactory final revisions. We look forward to seeing your revised manuscript and are grateful for your interest in The Journal of Physiology.

REFeree COMMENTS

Referee #1:

Reviewer's report

Authors have adequately responded to all my comments and have modified the text accordingly.

The data presented highlight an important aspect of skeletal muscle pathophysiology and should have a significant impact on our understanding of certain muscle disorders.

I have only two minor comments:

Fig. 1. Please explain why the colour-coded step patterns, i.e. the time during which a given paw is in contact with the glass, differs in panels C top right and C bottom right.

Fig. 2. It is not clear why there are no error bars in panel B.

Referee #2:

I still think that evaluation of the spindle function would have strongly strengthened the message (evaluation should not necessarily involve specific gamma input activation). The Ca²⁺ transients measurements add value to the work but do not prove that spindle function is impaired.

Thus, I concur with the author's statement to the Reviewing Editor that direct evidence that function impairment of the muscle spindles causes defects in gait and skeleton abnormalities is lacking.

Precise reference to which patient (number) in the Klein et al. (2012) paper and in the Zhou et al. 2007 paper, was used to design the dHT model should be given. Patient 35+ in the supplementary information of Klein et al. (c.14126C>T; p.Thr4709Met) does not seem to correspond to the dHT (p.Q1970fsX16+p.A4329D).

END OF COMMENTS

Dear Editors,

we were pleased to hear that our manuscript entitled “Massive reduction of RyR1 in muscle spindles of mice carrying recessive *Ryr1* mutations alters proprioception and causes scoliosis” by Ruiz et al., is acceptable for publication in the J. Physiology, provided we address the minor comments of the Editors and Reviewers.

We will reply to the comments on a point by point basis.

Reviewing Editor:

1. Please add a brief statement acknowledging the limitation that, while reduced Ca^{2+} transients suggest impaired spindle function, direct measurements of force generation or muscle contraction were not obtained.

Reply: We have added the sentence suggested by the Editor to the Conclusions section (page 28, lines 2-4, revised manuscript).

Senior Editor:

1. Please clarify the light/dark cycle for housing the animals.

Reply: We have added a sentence stating that mice were housed in cages and had a 12-h/12-h light/dark cycle, with lights on from 5:00 a.m. to 5:00 p.m. (page 7, lines 3, 4, revised manuscript).

Reviewer 1:

1. Fig. 1. Please explain why the colour-coded step patterns, differs in panels C top right and C bottom right.

Reply: We have amended the top panel C of Figure 1 so that now the same color-coded step patterns are present in panel C top and bottom.

2. It is not clear why there are no error bars in Fig.2 panel B.

Reply: The Relative change in RyR1 protein content between different tissues (liver, kidney, intrafusal and extrafusal muscles) was analyzed as described in the Methods section by comparing p-values and subsequently adjusting them for multiple test comparison, using the Benjamini-Hochberg method as implemented in the R/Bioconductor limma package. Using this type of analysis, there are no standard deviations. A sentence has been added on page 10, lines 21-23, revised manuscript.

Reviewer 2:

1. The Ca^{2+} transients measurements add value to the work but do not prove that spindle function is impaired. Thus, I concur with the author's statement to the Reviewing Editor that direct evidence that function impairment of the muscle spindles causes defects in gait and skeleton abnormalities is lacking.

Reply: See reply to Reviewing Editor above. In order to comply with the Reviewer and Reviewing Editor, we have added a sentence to the Conclusions section (page 28, lines 2-4, revised manuscript).

2. Precise reference to which patient (number) in the Klein et al. (2012) paper and in the Zhou et al. 2007 paper, was used to design the dHT model should be given. Patient 35+ in the supplementary information of Klein et al.

(c.14126C>T; p.Thr4709Met) does not seem to correspond to the dHT (p.Q1970fsX16+p.A4329D).

Reply: The Reviewer is correct; we wrote the wrong patient number in our previous reply. The dHT mice are isogenic to patient 56 (and not 35) in the paper by Klein et al, 2012, and patient 28 in the paper by Zhou et al, 2007. As requested, this information has been added to the revised manuscript, page 7, line 11 and page 17, lines 7-9).

We would like to thank the Editors and Reviewers for their constructive criticisms and hope that our paper is now accepted for publication in the Journal of Physiology.

Francesco Zorzato, M.D., PhD
Susan Treves, PhD

Dear Professor Treves,

Re: JP-RP-2025-287832R2 "**Massive reduction of RyR1 in muscle spindles of mice carrying recessive *Ryr1* mutations alters proprioception and causes scoliosis.**" by Alexis Ruiz, Sofia Benucci, Hervé Meier, Georg Schulz, Katarzyna Buczak, Christoph Handschin, Rodrigo C G Pena, Susan Treves, and Francesco Zorzato

We are pleased to tell you that your paper has been accepted for publication in The Journal of Physiology.

Yours sincerely,

Karyn Hamilton
Senior Editor
The Journal of Physiology

If you would like to receive our 'Research Roundup', a monthly newsletter highlighting the cutting-edge research published in The Physiological Society's family of journals (The Journal of Physiology, Experimental Physiology, Physiological Reports, The Journal of Nutritional Physiology and The Journal of Precision Medicine: Health and Disease), please click this link, fill in your name and email address and select 'Research Roundup':
<https://www.physoc.org/journals-and-media/membernews>

- You can help your research get the attention it deserves! Check out Wiley's free Promotion Guide for best-practice recommendations for promoting your work at: www.wileyauthors.com/eeo/guide. You can learn more about Wiley Editing Services which offers professional video, design, and writing services to create shareable video abstracts, infographics, conference posters, lay summaries, and research news stories for your research at: www.wileyauthors.com/eeo/promotion.

EDITOR COMMENTS

Reviewing Editor:

Nice study, no further comments.

Senior Editor:

Thank you for submitting your revised manuscript. We are pleased to accept it for publication in The Journal of Physiology.
Thank you for your interest in The Journal and Congratulations!